

# 1     Methane emissions from dairies in the Los Angeles Basin

Camille Viatte[1], Thomas Lauvaux[2], Jacob K. Hedelius[1], Harrison Parker[3], Jia Chen[4,*], Taylor Jones[4],
Jonathan E. Franklin[4], Aijun J. Deng[2], Brian Gaudet[2], Kristal Verhulst[5], Riley Duren[5], Debra
Wunch[1,**], Coleen Roehl[1], Manvendra K. Dubey[2], Steve Wofsy[4], and Paul O. Wennberg[1]
[1] Division of Geological and Planetary Sciences, California Institute of Technology, Pasadena, CA,
US
[2] Department of Meteorology, Pennsylvania State University, University Park, PA, US
[3] Earth System Observations, Los Alamos National Laboratory, Los Alamos, NM, US
[4] School of Engineering and Applied Sciences, Harvard University, Cambridge, MA, US
[5] Jet Propulsion Laboratory, California Institute of Technology, Pasadena, California, US
[*] Now at Electrical, Electronic and Computer Engineering, Technische Universtat of Munich,
Munich, Germany
[**] Now at Department of Physics, University of Toronto, Toronto, ON, Canada



Abstract
We estimate the amount of methane ($CH_4$) emitted by the largest dairies in the southern
California region by combining measurements from four mobile solar-viewing ground-based
spectrometers (EM27/SUN), in situ isotopic $^{13/12}CH_4$ measurements from a CRDS analyzer
(Picarro), and a high-resolution atmospheric transport simulations with Weather Research and
Forecasting model in Large-Eddy Simulation mode (WRF-LES).
The remote sensing spectrometers measure the total column-averaged dry-air mole fractions of
$CH_4$ and $CO_2$ ($X_{CH4}$ and $X_{CO2}$) in the near infrared region, providing information about total
emissions of the dairies at Chino. Gradients measured by the four EM27/SUN ranged from 0.2 to
22 ppb and from 0.7 to 3 ppm for $X_{CH4}$ and $X_{CO2}$, respectively. To assess the fluxes of the dairies,
these gradient measurements are used in conjunction with the local atmospheric dynamics from
wind measurements at two local airports and from the WRF-LES simulations at 111 m resolution.
Our top-down $CH_4$ emissions derived using the Fourier Transform Spectrometers (FTS)
observations of 1.4 to 4.8 ppt/s are in the low-end of previous top-down estimates, consistent
with reductions of the dairy farms with urbanization in the domain. However, the wide range of
inferred fluxes points to the challenges posed by heterogeneity of the sources and meteorology.
Inverse modeling from WRF-LES is utilized to resolve the spatial distribution of $CH_4$ emissions in
the domain. Both the model and the measurements indicate heterogeneous emissions, with
contributions from anthropogenic and biogenic sources at Chino. A Bayesian inversion and a
Monte-Carlo approach are used to provide the $CH_4$ emissions of 3.2 to 4.7 ppt/s at Chino.



## 1) Introduction

Atmospheric methane ($CH_4$) concentration has increased by 150% since the pre-industrial era, contributing to a global average change in radiative forcing of 0.5 $W.m^{-2}$ (Foster et al., 2007; Myhre et al., 2013). Methane is naturally emitted by wetlands, but anthropogenic emissions now contribute more than half of its total budget (Ciais et al., 2013), ranking it the second most important anthropogenic greenhouses gas after carbon dioxide ($CO_2$).

The United Nations Framework Convention on Climate Change (UNFCCC, http://newsroom.unfccc.int/) aims to reduce this $CH_4$ emission by reaching global agreements and collective action plans. In the United States (US), the federal government aims to reduce $CH_4$ emissions by at least 17% below 2005 levels by 2020 by targeting numerous key sources such as (in order of importance): agriculture, energy sectors (including oil, natural gas, and coal mines), and landfills (Climate Action Plan, March 2014). Methane emissions are quantified using "bottom-up" and "top down" estimates. The "bottom-up" estimates are based on scaling individual emissions and process level information statistically (such as the number of cows, population density or emission factor) with inherent approximations. "Top-down" estimates, based on atmospheric $CH_4$ measurements, often differ from these reported inventories both in the total emissions and the partitioning among the different sectors and sources (e.g. Hiller et al., 2014). In the US, the disagreement in $CH_4$ emissions estimated can reach a factor of two or more (Miller et al., 2013; Kort et al., 2014), and remains controversial regarding the magnitude of emissions from the agricultural sector (Histov et al. 2014). Thus, there is an acknowledged need for more accurate atmospheric measurements to verify the bottom-up estimates (Nisbet and Weiss, 2010). This is especially true in urban regions, such as the Los Angeles basin, where many different $CH_4$ sources (from farm lands, landfills, and energy sectors) are confined to a relatively small area of ~87000 $km^2$ (Wunch et al., 2009; Hsu et al. 2010; Wennberg et al., 2012; Peischl et al., 2013; Guha et al., 2015; Wong et al., 2015). Therefore, flux estimation at a local scales is needed to resolve discrepancies between bottom-up and top-down approach and improve apportion among $CH_4$ sources.



Inventories of $CH_4$ fluxes suggest that emissions from US agriculture increased by more than 10%
between 1990 and 2013 (Environmental Protection Agency, EPA, 2015), and by more than 20%
since 2000 in California (California Air Resources Board, CARB, 2015). In addition, these emissions
are projected to increase globally in the future due to increased food production (Tilman and
Clark, 2014). Livestock in California have been estimated to account for 63% of the total
agricultural emissions of greenhouse gases (mainly $CH_4$ and $N_2O$); dairy cows represented more
than 70% of the total $CH_4$ emissions from the agricultural sectors in 2013 (CARB, 2015). State-
wide actions are now underway to reduce $CH_4$ emissions from dairies (ARB concept paper, 2015).
Measurements at the local-scale with high spatial- and temporal-resolution are needed to assess
$CH_4$ fluxes associated with dairy cows and to evaluate the effectiveness of changing practices to
mitigate $CH_4$ emissions from agriculture.
Space-based measurements provide the dense and continuous datasets needed to constrain $CH_4$
emissions through inverse modeling (Streets et al., 2013). Recent studies have used the
Greenhouse gases Observing SATellite (GOSAT – footprint of ~10 km diameter) observations to
quantify mesoscale natural and anthropogenic $CH_4$ fluxes in Eurasia (Berchet et al., 2015) and in
the US (Turner et al., 2015). However, it is challenging to estimate $CH_4$ fluxes at smaller spatial
scales using satellite measurements due to their large observational footprint (Bréon and Ciais,
2010). Nevertheless, recent studies used the SCanning Imaging Absorption spectroMeter for
Atmospheric CHartographY (SCIAMACHY – footprint of 60 km x 30 km) to assess emissions of a
large $CH_4$ point source in the US (Leifer et al., 2013; Kort et al., 2014).
Small-scale $CH_4$ fluxes are often derived from in situ measurements performed at the surface and
from towers (Zhao et al., 2009), and/or in situ and remote-sensing measurements aboard aircraft
(Karion et al.,2013; Peischl et al., 2013; Lavoie  et al., 2015; Gordon et al., 2015). A recent study
emphasized the relatively large uncertainties of flux estimates from aircraft measurements using
the mass balance approach in an urban area (Cambaliza et al., 2014).
Ground-based solar absorption spectrometers are powerful tools that can be used to assess local
emissions (McKain et al., 2012). This technique has been used to quantify emissions from regional





90 to urban scales (Wunch et al., 2009; Stremme et al., 2013; Kort et al., 2014; Lindenmaier et al.,

91 2014; Hase et al., 2015; Franco et al., 2015, Wong et al., 2015, Chen et al., 2016).

92 In this study, we use four mobile ground-based total column spectrometers (called EM27/SUN,

93 Gisi et al., 2012) to estimate $CH_4$ fluxes from the largest dairy-farming area in the South Coast Air

94 Basin (SoCAB), located in the city of Chino, in San Bernardino County, California. The Chino area

95 was once home to one of the largest concentrations of dairy farms in the United States (US),

96 however rapid land-use change in this area may have caused $CH_4$ fluxes from the dairy farms

97 change rapidly in both space and time. Chen et al. (2016) used differential column measurements

98 (downwind minus upwind column gradient $\Delta X_{CH4}$ across Chino) on a high-wind data to verify

99 emissions reported in the literature. In this study, the same column measurement network is

100 employed in conjunction with meteorological data and a high-resolution model to estimate $CH_4$

101 emissions at Chino for several different days, including more typical wind conditions.

102 In section 2 of this paper, the January 2015 field campaign at Chino is described, with details

103 about the mobile column and in situ measurements. In section 3, we describe the new high

104 resolution Weather Research & Forecasting (WRF) model with Large Eddy Simulations (LES)

105 setup. In section 4, results of $CH_4$ fluxes estimates are examined. Limitations of this approach, as

106 well as suggested future analyses are outlined in section 5.





2)   Measurements in the Los Angeles Basin dairy farms
2.1) Location of the farms: Chino, California
Chino (34.02°N, -117.69°W) is located in the eastern part SoCAB, called the Inland Empire, and
has historically been a major center for dairy production. With a growing population and
expanding housing demand, the agricultural industry has shrunk in this region and grown in the
San Joaquin Valley (California Central Valley). The number of dairies decreased from ~400 in the
1980's to 95 in 2013 (red area of panels a, b, and c in Figure 1). Nevertheless, in 2013 ~90 % of
the southern California dairy cow population (California Agricultural Statistics, 2013) remained
within the Chino area of ~6 x 9 km (Figure 1). These feedlots are a major point source of $CH_4$ in
the Los Angeles basin (Peischl et al., 2013).
2.2) Mobile column measurements: EM27/SUN
Atmospheric dry-air mole fractions of $CH_4$ and $CO_2$ (denoted $X_{CH4}$ and $X_{CO2}$, Wunch at al., 2011)
have been measured using four ground-based mobile Fourier Transform Spectrometers (FTS).
The mobile instruments were purchased from Bruker Optics, are all EM27/SUN models. The four
FTS (two owned by Harvard University, denoted Harvard 1 and 2, one owned by Los Alamos
National Laboratory, denoted LANL, and one owned by the California Institute of Technology,
denoted Caltech, were initially gathered at the California Institute of Technology in Pasadena,
California in order to compare them against the existing Total Carbon Column Observing Network
(TCCON, Wunch et al., 2011) station and to each other, over several full days of observation. The
instruments were then deployed to Chino to develop a methodology to estimate greenhouses
gas emissions and improve the uncertainties on flux estimates from this major local source.
Descriptions of the capacities and limitations of the mobile EM27/SUN instruments have been
published in Chen et al. (2016) and Hedelius et al. (2016). For this analysis, we need to ensure
that all the data from the EM27/SUN instruments are on the same scale. Here, we reference all
instruments to the Harvard2 instrument. Standardized approaches (retrieval consistency,
calibrations between the instruments) are needed to monitor small atmospheric gradients using
total column measurements from the EM27/SUN. Indeed we ensured all retrievals used the same



algorithm, calibrated pressure sensors, and were scaled according to observed, small systematic
differences to reduce instrumental biases (Hedelius et al., 2016).
These modest resolution (0.5 cm$^{-1}$) spectrometers are equipped with solar-trackers (Gisi et al.,
2011) and measure throughout the day. To retrieve atmospheric total column abundances of
$CH_4$, $CO_2$, and oxygen ($O_2$) from these Near InfraRed (NIR) solar absorption spectra, we used the
GGG software suite, version GGG2014 (Wunch et al., 2015). Column measurements at Chino
were obtained on five days: the 15$^{th}$, 16$^{th}$, 22$^{nd}$ and 24$^{th}$ of January, and the 13$^{th}$ of August, 2015.
Of these days, January 15$^{th}$, 16$^{th}$, and 24$^{th}$ are sufficiently cloud-free for analysis. These days have
different meteorological conditions (i.e. various air temperatures, pressures, wind speeds and
directions), improving the representativeness of the flux estimates at Chino.
Figure 1 shows measurements made on January 15$^{th}$, 16$^{th}$, and 24$^{th}$. Wind speeds and directions,
shown in the bottom panels of Figure 1, are measured at the two local airports inside the domain
(the Chino airport indicated on panels d, e, and f and the Ontario airport on panels g, h, and i).
Wind measurements from these two airports, located at less than 10 km apart, are made at an
altitude of 10 meters above the surface. The exact locations of the four EM27/SUN (colored
symbols in Figure 1 in the upper panels a, b, and c) were chosen each morning of the field
campaign to optimize the chance of measuring upwind and downwind of the plume. On the 15$^{th}$
and 16$^{th}$ of January, the wind speed was low with a maximum of 3 ms$^{-1}$ and highly variable
direction all day (Figure 1, panels d, e, g and h), therefore the four EM27/SUN were placed at
each corner of the source area to ensure that the plume was detected by at least one of the
instruments throughout the day. On the contrary, the wind in January 24$^{th}$ had a constant
direction from the Northeast and was a relatively strong 8-10 ms$^{-1}$ (Figure 1, panels f and i), so
the instruments were located such that one spectrometer (Harvard2) was always upwind (blue
symbols in Figure 1) and the others are downwind of the plume and at different distances from
the sources (black, green, and red symbols in Figure 1).

2.3) In situ measurements: Picarro

The EM27/SUN column measurements are supplemented by ground-based in situ measurement
using a commercial Picarro instruments during January campaign. The Picarro instruments use a



Cavity Ringdown Spectroscopy (CRDS) technique that employs a wavelength monitor and
attenuation to characterize species abundance.
In situ $^{12}CH_4$, $CO_2$, and $^{13}CH_4$ measurements were performed on January 15[th], 16[th], and 22[nd], and
August 13[th] 2015 at roughly 2m away from the LANL EM27/SUN (Figure 1 a, b, and c) with a
Picarro          G2132-I          instrument          (Arata          et          al.,          2016,
http://www.picarro.com/products_solutions/isotope_analyzers/). This Picarro, owned by LANL,
utilize a 1/4" synflex inlet tube placed approximately 3m above ground level to sample air using
a small vacuum pump. Precisions on $^{12}CH_4$, $CO_2$, and $^{13}CH_4$ measurements are 6 ppb, 2 ppm, and
0.6 ‰, respectively.
To locate the major $CH_4$ sources in the dairy farms area, a second Picarro G2401 instrument
(http://www.picarro.com/products_solutions/trace_gas_analyzers/) from the Jet Propulsion
Laboratory (JPL) was deployed on January 15[th], 2015. Precision on $CH_4$ measurements is ~1 ppb.



3) Model simulations

3.1) Description of WRF-LES model

The Weather Research and Forecasting (WRF) model (Skamarock et al., 2008) is an atmospheric
dynamics model used for both operational weather forecasting, and scientific research
throughout the global community. Two key modules that supplement the baseline WRF system
are used here. First, the chemistry module WRF-Chem (Grell et al., 2005) adds the capability of
simulating atmospheric chemistry among various suites of gaseous and aerosol species. In this
study, $CH_4$ is modeled as a passive tracer because of its long life time relative to the advection
time at local scales. The longest travel time from the emission source region to the instrument
locations is less than an hour, which is extremely short compared to the lifetime of $CH_4$ in the
troposphere (~9 years). Therefore, no specific chemistry module is required. The version of WRF-
Chem used here (Lauvaux et al., 2012) allowed for the offline coupling between the surface
emissions, prescribed prior to the simulation, and its associated atmospheric tracers. Second, we
make use of the Large Eddy Simulation (LES) version of WRF (Moeng et al., 2007) on a high-
resolution model grid with 111-m horizontal grid spacing. A key feature of the simulation is the
explicit representation of the largest turbulent eddies of the Planetary Boundary Layer (PBL) in a
realistic manner. The more typical configuration of WRF (and other atmospheric models) is to be
run at a somewhat coarser resolution that is incapable of resolving PBL eddies. An advantage in
this study is that the integrated effect of all PBL eddies on vertical turbulent transport is
parameterized. By having a configuration with the combination of $CH_4$ tracers and PBL eddies,
we can realistically predict the evolution of released material at scales on the order of the PBL
depth or smaller.
In this real case experiment, the model configuration consists of a series of four one-way nested
grids, shown in Figure 2 and described further in the supplementary information section (S1).
Each domain contains 201 x 201 mass points in the horizontal, with 59 levels from the surface to
50 hPa, and the horizontal grid spacings are 3 km, 1 km, 333 m, and 111 m. All four domains use
the WRF-Chem configuration. The model 3-km, 1-km, and 333-m grids are run in the conventional
mesoscale configuration with a PBL parameterization, whereas the 111-m grid physics is LES. The





initial conditions for the cases are derived from the National Centers for Environmental
Prediction (NCEP) 0.25-degree Global Forecasting System (GFS) analysis fields (i.e., 0-hour
forecast) at 6-hour intervals. The simulations are performed from 12:00 to 00:00 UTC (= 04:00 to
16:00 LT) only, which corresponds to daylight hours when solar heating of the surface is present
and measurements are made.
Data assimilation is performed using Four Dimensional Data Assimilation (FDDA; Deng et al.,
2009) for the 3-km and 1-km domains. The assimilation improves the model performance
significantly (Rogers et al., 2013) without interfering with mass conservation and the continuity
of the air flow.  Surface wind and temperature measurements, including from the Ontario (KONT)
and Chino (KCNO) airport stations, and upper-air measurements were assimilated within the
coarser grids using the WRF-FDDA system. However, no observations of any kind were
assimilated within the 333-m and 111-m domains; therefore, the influence of observations can
only come into these two domains through the boundary between the 333-m and 1-km grids.
Wind measurements at fine scale begin to resolve the turbulent perturbations, which would
require an additional pre-filtering. These measurements are used to evaluate the WRF model
performances at high resolutions.
Based on the terrain elevation in the LES domain (Figure 2), target emissions are located in a
triangular-shaped valley with the elevation decreasing gradually towards the South. However,
hills nearly surround the valley along the southern perimeter. Meanwhile, the foothills of the San
Gabriel Mountains begin just off the 111-m domain boundary to the North. As a result, the wind
fields in the valley are strongly modified by local topography, and can be quite different near the
surface than at higher levels.

3.2) Atmospheric inversion methodology: Bayesian framework and Simulated

Annealing error assessment

Due to the absence of an adjoint model in Large Eddy Simulation mode, the inverse problem is
approached with Green's functions, which correspond to the convolution of the Chino dairies
emissions and the WRF-LES model response. For the two simulations (January 15[th] and 16[th]), 16
rectangular areas of 2 x 2 km$^2$ (Figure 2) are defined across the feedlots to represent the state



vector ($x$) and therefore the spatial resolution of the inverse emissions, which correspond to the
entire dairy farms area of about 8 x 8 km$^2$ once combined together. The 16 emitting areas
continuously release a known number of CH$_4$ molecules (prior estimate) during the entire
simulations, along with 16 individual tracers representing the 16 areas of the dairies area. The
final relationship between each emitting grid-cell and each individual measurement location is
the solution to the differential equation representing the sensitivity of each column
measurement to the different 2 x 2 km$^2$ areas. The WRF-LES results are sampled every 10 minutes
at each sampling location to match the exact measurement times and locations of the EM27/SUN
instruments.
The inversion of the emissions over Chino is performed using a Bayesian analytical framework,
described by the following equation:

$$x = x_0 + BH^T(HBH^T + R)^{-1}(y - Hx_0) \qquad (1)$$

with $x$ the inverse emissions, $x_0$ the prior emissions, $B$ the prior emission error covariance, $R$ the
observation error covariance, $H$ the Green's functions, and $y$ the observed column dry air mole
fractions. The dimension of the state vector is 16, assuming constant CH$_4$ emissions for each
individual day. Two maps of 16 emission estimates are produced corresponding to the 2 x 2 km$^2$
areas for the two days (January 15$^{th}$ and 16$^{th}$). A combined inversion provides a third estimate of
the emissions using 10-minute average column data from both days. The definition of the prior
error covariance matrix $B$ is most problematic because little is known about the dairy farms
emissions except the presence of cows distributed in lots of small sizes. However, we assumed
no error correlation as it is known that cows are not distributed randomly across Chino. For the
definition of the variances in $B$, no reliable error estimate is available. The lack of error estimate
impacts directly the inverse emissions, therefore results in the generation of unreliable posterior
error estimates. Instead, we develop a Monte-Carlo approach using a Simulated Annealing (SA)
technique. We test the initial errors in the emissions by creating random draws with an error of
about 200% compared to the expected emissions (based on the dairy cows' emissions from CARB
2015). We then generate populations of random solutions and iterated 2000 times with the
Simulated Annealing algorithm. The metric used to select the best solutions is the Mean Absolute



Error (or absolute differences) between the simulated and observed column fractions. We store
the solutions exhibiting a final mismatch of less than 0.01 ppm to minimize the mismatch
between observed and simulated column fractions. The optimal solution and the range of
accepted emission scenarios are shown in Figure S2. The space of solutions provide a range of
accepted emissions for each 2 x 2 km$^2$ area that can be used as a confidence interval in the
inversion results. The posterior emissions from the Bayesian inversion are then compared to the
confidence interval from the Simulated Annealing to evaluate our final inverse emissions
estimates and the posterior uncertainties. The results are presented in Section 4.3.



4) Results

4.1) Observations of $X_{CH4}$ and $X_{CO2}$ in the dairy farms

Figure 3 shows the 1-minute average time series of $X_{CH4}$ (upper panels a, b, and c) and $X_{CO2}$ (d, e,
and f) derived from the four EM27/SUN. For days with slow wind, i.e. on January 15th and 16th
(Figure 1, panels d, e, g and h), the maximum gradients observed between the instruments are
17 and 22 ppb (parts per billion), and 2 and 3 ppm (parts per million), for $X_{CH4}$ and $X_{CO2}$,
respectively. Assuming that the observed Xgas changes are confined to the PBL, gradients in this
layer are about ten times larger. Gradients observed on January 15th and 16th are higher than
those of $X_{CH4}$ and $X_{CO2}$ of 2 ppb and 0.7 ppm observed on a windy day, the 24th. The $X_{CH4}$ and $X_{CO2}$
variabilities captured by the instruments are due to changes in wind speed and direction, i.e.,
with high $X_{CH4}$ signals when the wind blows from the dairies to the instruments. Thus, the
EM27/SUN are clearly able to detect variability of greenhouses gases at local scales (temporal:
less than 5 minutes, and spatial: less than 10 km) indicating that these mobile column
measurements have the potential to provide estimates of local source emissions.

4.2) Estimation of fluxes with EM27/SUN column measurements

Total column measurements are directly linked to total emissions (McKain et al., 2012) and are
sensitive to surface fluxes (Keppel-Aleks et al., 2012). To derive the total emissions of trace gases
released in the atmosphere from a source region, the "mass balance" approach is often used. In
its simplest form, the $X_{CH4}$ fluxes can be written as in Equation 2, but this requires making
assumptions about the homogeneity of the sources and wind shear in the PBL.
$$F_{X_{CH4}} = \Delta_{X_{CH4}} \frac{V(z)}{m(\theta)} C_{air}(z) \qquad (2)$$
where $F_{X_{CH4}}$ is the flux (molecules/s.m²), $\Delta_{X_{CH4}}$ is the $X_{CH4}$ enhancement between the upwind and
the downwind region (ppb), $V$ is the average wind speed (ms⁻¹) from both airports, m is the
distance in meter that air crosses over the dairies calculated as a function of the wind direction
$\theta$, and $SC_{air}(z)$ is the vertical column density of air (molecules/m²). The distances that airmasses
cross over the dairies (m) before reaching a receptor (EM27/SUN) are computed for each day,
each wind direction, and each instrument (see complementary information section S3).





Equation 2 can be reformulated as:
$$\Delta_{X_{CH4}} = \Delta_t \cdot \frac{F_{X_{CH4}}}{C_{air(z)}} \qquad (3)$$

where $\Delta t = \frac{m(\theta)}{V(z)}$ is the residence time of air over the dairies (in seconds).
A modified version of this mass balance approach has been used by Chen et al. (2016) to verify
that the $X_{CH4}$ gradients measured by the EM27/SUN are comparable to the expected values
measured at Chino during the CalNex aircraft campaign (Peischl et al., 2013). In Chen et al., $X_{CH4}$
enhancements measured between upwind and two of the downwind sites on January 24[th] (day
of constant wind direction, Figure 1 panels f and i) are compared to the expected value derived
from Peischl's emission numbers, which were determined using the bottom-up method and
aircraft measurements. They found that the measured $X_{CH4}$ gradient of ~2 ppb, agrees within the
low range of the 2010 value. However, this differential approach, using upwind and downwind
measurements, reduces the flux estimates to only one day (January 24[th]), since the wind speed
and direction were not constant during the other days of field measurements.
In this study, we extend the analysis of the Chino dataset using the mass balance approach on
steady-wind day (on January 24[th]) for all the FTS instruments (i.e three downwind sites), as well
as employing the other two days of measurements (January 15[th], and 16[th]) in conjunction the
WRF-LES model to derive a flux of $X_{CH4}$ from the dairy farms. We exclude measurements from
January 22[nd] and August 13[th] because of the presence of cirrus clouds during those days, which
greatly reduce the precision of the column measurements. Our $X_{CH4}$ signal measured by the FTS
can be decomposed as the sum of the background concentration and the enhancements due to
the local sources:
$$X_{CH4,measured} = X_{CH4,background} + \Delta_{X_{CH4}} \quad (4)$$

Gradients of $X_{CH4}$ ($\Delta_{X_{CH4}}$) are calculated relative to one instrument for the three days. The $X_{CH4}$
means (and standard deviations) over the three days of measurements at Chino are 1.824
(±0.003) ppm, 1.833 (±0.007) ppm, 1.823 (±0.003) ppm, and 1.835 (±0.010) ppm for the Caltech,
Harvard1, Harvard2, and LANL instruments, respectively. The Harvard2 $X_{CH4}$ mean and standard





deviation are the lowest of all the observations, therefore these measurements are used to
calculate gradients of $X_{CH4}$ ($\Delta_{X_{CH4}}$) for the other three instruments. Gradients of $X_{CH4}$ ($\Delta_{X_{CH4}}$) for an
instrument $i$ (i.e. Caltech, Harvard1, or LANL) are the differences between each 10-minute
average $X_{CH4}$ measured by $i$ and the corresponding 10-minute average $X_{CH4}$ measured by the
Harvard2 instrument. Details about the residence time calculation can be found in the
supplementary information section (S3). Time series of anomalies for individual measurements
days are presented in Figure 4.
Assuming the background levels $X_{CH4}$ are similar at all the instrument sites within 10 km distance
and steady state wind fields, equation 3 can be written as:
$$(X_{CH4,i} - X_{CH4,Harvard2}) \propto (t_i - t_{Harvard2}).F_{X_{CH4}} \quad (5)$$
Graphical representation of equation 5 is shown in Figure 5 in which $\Delta_{X_{CH4}}$, the measured
gradients by the four FTS during January 24$^{th}$, is plotted as a function of $\Delta_t$, so that the slope
corresponds to a flux in ppb/s or ppt/s (parts per trillion). In this figure the slope of the blue lines
(dark and light ones) represents the flux measured at Chino in previous studies (Peischl et al.,
2013). These studies estimating CH4 fluxes at Chino in 2010 reported a bottom-up value of 28
Gg/yr with a range of top-down measurements from 24 to 74 Gg/yr (Table 1). To compare these
values (in Gg/yr) to the fluxes derived from column average (in ppt/s), we used Equation 6:
$$F_{col} = \frac{F.10^9}{a.Y.C_{air}(z).\frac{m_g}{Na}} . 10^{12} \quad (6)$$
where $F_{col}$ is the column average flux in ppt/s, $F$ the flux in Gg/yr, $a$ the area of Chino (m) , $Y$
the number of seconds in a year, $C_{air}(z)$ the vertical column density of air (molecules/m²), $m_g$
the molar mass of CH4 (g/mol), and $Na$ the Avogadro constant (mol⁻¹).
On January 24$^{th}$, when the wind speed is higher than the other days (Figure 1, panels f, and i), the
residence time over the dairies ($\Delta_t$) is reduced by a factor of 30. The mean $\Delta_t$ from the closest to
the furthest instruments to the upwind site are 4 minutes for Caltech (black square, Figure 5), 13
minutes for Harvard2 (green square, Figure 5), and 16 minutes for LANL (red square, Figure 5).
The $X_{CH4}$ fluxes estimated using the mean states (mass balance approach) are 4.8, 1.6, and 1.4



ppt/s for the Caltech, LANL, and Harvard2 downwind instruments. Overall, the FTS network infers
$X_{CH4}$ emissions at Chino that are in the low-end of previous top-down estimates reported by
Peischl et al. (2013), which is consistent with the decrease in cows and farms in the Chino area
over several past years.
However, the flux estimated using the closest instrument/shortest residence time (i.e. Caltech)
exceeds the value from previous studies by almost a factor of two. The other values from LANL
and Harvard2, on the other hand, are lower than previous published values. This analysis
demonstrates that, even with the steady-state winds day, and the simple geometry, the mass
balance still have weaknesses, since it does not properly explain the differences seen among the
three downwind sites. The close-in site exhibits the highest apparent emission rate possibly due
to the proximity of a large $CH_4$ source. This exhibits delusive approximations implied by this
method (i.e., spatial inhomogeneity of $X_{CH4}$ sources completely averaged out and conservative
transport in the domain) even on "golden day" of strong steady-state wind pattern. Therefore,
when investigating emissions at local scales these assumptions can be dubious and lead to errors
in the flux estimates.

4.3) Spatial study of the $CH_4$ fluxes using WRF-LES data

Analysis of the spatial sources at Chino is developed in this section using the WRF-LES model and
in section 4.4 with in situ Picarro measurements.
To map the sources of $CH_4$ at Chino with the model, we focus on the two days of measurements
during which the wind changed direction regularly (i.e. January 15$^{th}$ and 16$^{th}$, Figure 1 panels d,
e, g and h). This provides the model information about the spatial distribution of $CH_4$ emissions.

4.3.1) WRF-LES model evaluation

The two WRF-Chem simulations were evaluated for both days (January 15$^{th}$ and 16$^{th}$) using
meteorological observations (Figures 6 and 7). Starting with the larger region on the 3-km grid
where WMO sondes are available (Figure 6), model verification for both days indicates that wind
speed errors averaged over the domain are only about 1 ms$^{-1}$ in the free atmosphere and slightly
larger in the PBL (less than 2 ms$^{-1}$). For wind direction, the Mean Absolute Error (MAE) is less than





20 degrees in the free atmosphere and increases approaching the surface, reaching a maximum
of about 50 degrees there. More relevant to this study, the Mean Error (ME) remains small over
the profile and more specifically in the PBL, oscillating between 0 and 10 degrees. At higher
resolutions, the comparison between observed and WRF-predicted surface wind speed (Figure
7) indicates that WRF is able to reproduce the overall calm wind conditions for both days at both
WMO stations, Chino (KCNO) and Ontario (KONT). However, measurements below 1.5 ms$^{-1}$ are
not reported following the WMO standards, which limits the ability to evaluate the model over
time. On January 15$^{th}$ at KCNO, consistent with the observations, all domains except the 3-km
grid predict no surface wind speeds above 2 ms$^{-1}$ from 16:00 – 19:00 UTC, except for one time
from the 111-m LES domain. After this period, the 111-m LES domain successfully reproduces the
afternoon peak in wind speed of about 3 ms$^{-1}$, only slightly larger than the observed values (3.6
ms$^{-1}$ at Chino and 3.9 ms$^{-1}$ at Ontario airports). However, we should not expect perfect
correspondence between the observations and the instantaneous LES output unless a low-pass
filter is performed on the LES to average out the turbulence.  On January 16$^{th}$ 2015, the model
wind speed at KONT remained low throughout the day, in good agreement with the (unreported)
measurements, and also with available observations.
4.3.2) Dispersion of tracers in LES mode: 15$^{th}$ and 16$^{th}$ January 2015
We use the January 15$^{th}$ 2015 case as an example showing the detail in the local winds that can
be provided by the high-resolution LES domain. Prior to approximately 19:00 UTC (= 11:00 LT) a
brisk easterly flow is present in the valley up to a height of 2 km; however, near the surface, a
cold pool up to several hundred meters thick developed with only a very weak easterly motion.
A simulated tracer released from a location near the east edge of the Chino area stays confined
to the cold pool for this period (Figure 8, upper row). Solar heating causes the cold pool to break
down quite rapidly after 19:00 UTC, causing the low-level wind speed to become more uniform
with height (around 3 ms$^{-1}$ from the east), and allowing the tracer to mix up to a height of about
1 km (Figure 8, middle row). Beginning around 22:00 UTC (= 14:00 LT) however, a pulse of easterly
flow scours out the valley from the east, while a surge of cooler westerly flow approaches at low
levels from the west, undercutting the easterly flow. By 00:00 UTC (=16:00 LT) the tracer seems



to be concentrated in the cooler air just beneath the boundary of the two opposing air streams
(Figure 8, lower row).
The tracer released (right columns in Figure 8) from an emitting 2 x 2 km$^2$ pixel shows complex
vertical structures and two different regimes over the day. At 18:00 UTC, the tracer is
concentrated near the surface, except toward the West with a maximum at 600 m high. At 21:00
UTC, the tracer is well-mixed in the vertical across the entire PBL, from 0 to about ~1 km,
corresponding to convective conditions of daytime. At 00:00 UTC, the stability increased again,
generating a low vertical plume extent with complex structures and large vertical gradients along
the transect. Several updrafts and downdrafts are visible at 18:00 and 00:00 UTC, indicated by
the shift in wind vectors and the distribution of the tracer in the vertical (Figure 8). These spatial
structures are unique to the LES simulation, as the PBL scheme of the mesoscale model does not
reproduce turbulent eddies within the PBL.
In the horizontal, convective rolls and large tracer gradients are present, with visible fine-scale
spatial structures driven by the topography (i.e. hills in the South of the domain) and turbulent
eddies. Figure 9 (left panel) illustrates the spatial distribution of the mean horizontal wind at the
surface over the 111-m simulation domain at 18:00 UTC, just prior to the scouring out of the cold
pool near a large Chino feedlot. It can be seen that the near-surface air that fills the triangular
valley in the greater Chino area is nearly stagnant, while much stronger winds appear on the
ridges to the south. There are some banded structures showing increased wind speed near KONT
to the north of the main pool of stagnant air. Figure 9 (right panel) illustrates the wind pattern
for the 18:00 UTC January 16[th] case. The same general patterns can be seen, with the main
apparent differences being reduced wind speed along the southern high ridges, and more
stagnant air in the vicinity of KONT along with elevated wind speed bands near KCNO. These
results emphasize how variable the wind field structures can be from point-to-point in the valley.
4.3.3) Bayesian inversion and error assessment
We present the inverse emissions from the Bayesian analytical framework in Figures 10. The
Bayesian analytical solution was computed for both days, assuming a flat prior emission rate of
2150 mol/km$^2$/hour corresponding to a uniform distribution of 115000 dairy cows over 64 km$^2$





emitting methane at a constant rate of 150 kg of $CH_4$ per year (CARB 2015), plus 18 kg annually
per cow from dry manure management assumed to be on-site (Peischl et al., 2013). The colored
areas in Figure 10 represent the ranges of solutions defined by the Simulated Annealing (SA)
analysis, for the two days of the campaign (in blue and green). The Bayesian averages agree well
with the SA estimates, with high confidence for half of the pixels (1, 2, 3, 4, 8, 13, 15, and 16),
and lower confidence for the other pixels. High values coincide with high confidence, which
confirms the fact that large signals constrain the inverse solution better. This would possibly
suggest that only the largest emissions could be attributed with sufficient confidence using these
tools.
The spatial distribution of the emissions is shown in Figure 13, which directly corresponds to the
pixel emissions presented in Figure 10. The largest sources are located in the southern part of
the dairy farms area, and in the northeastern corner of the domain. Additional interpretation of
these results is presented in the following section. The combination of the results from two dates
(January 15$^{th}$ and 16$^{th}$) is necessary in order to identify the whole southern edge of the feedlots
as a large source. Sensitivity results are presented in the discussion and in the supplementary
information section (S4 and S5). Additional sensitivity tests were performed to evaluate the
impact of instrument errors, introducing a systematic error of 5 ppb in $X_{CH4}$ measured by one of
the EM27/SUN. The posterior emissions increased by 3-4 Gg/year for a +5ppb bias almost
independent of the location of the biased instrument. This represents ~10% of the total emission
at Chino.

4.4) Spatial study of the $CH_4$ emissions at Chino using Picarro measurements

During the field campaign in January 2015, in situ measurements of $CH_4$, $CO_2$, as well as $\delta^{13}C$ are
performed simultaneously with a Picarro instrument at the same site as the LANL EM27/SUN.
Fossil-related $CH_4$ sources, such as power plants, traffic, and natural gas, emit $CH_4$ with an isotopic
depletion $\delta^{13}C$ ranging from -30 to -45 ‰, whereas biogenic methane sources, such as those from
enteric fermentation and wet and dry manure management in dairies and feedlots emit in the
range of –65 to –45 ‰ (Townsend-Small et al., 2012). During the January 2015 campaign, the $\delta^{13}C$
at Chino ranged from -35 to -50 ‰, indicating a mixture of fossil and biogenic sources





respectively. Most of the air sampled included a mixture of both sources. However, the
measurements with the highest $CH_4$ concentrations had lowest $\delta^{13}C$ signatures, suggesting that
the major $CH_4$ enhancements can be attributed to the dairy farms and not the surrounding urban
sources.
On January 16th and 22nd, the Picarro and the LANL EM27/SUN were installed at the southwest
side of the largest dairies in Chino (red pin, Figure 1b), near a wet lagoon that is used for manure
management (< 150 m away). For these days, the Picarro measured enhancements of $CH_4$ up to
20 ppm above background concentrations, demonstrating that the lagoon is a large source of $CH_4$
emissions in the Chino area. The location of the lagoon was identified and verified by satellite
imagery, visual inspection, and also with measurements from the second Picarro instrument
deployed in the field on January 15th, 2015.  With this instrument, $CH_4$ spikes up to 23 ppm were
observed near the wet manure lagoon. The measurements from both Picarros and the LANL
EM27/SUN instrument near the lagoon suggested that this is a significant local source of $CH_4$
emissions in the Chino area.
As opposed to column measurements, Picarro measurements are very sensitive to the dilution
effect of gases in the PBL. With a low boundary layer, atmospheric constituents are concentrated
near the surface, and the atmospheric signal detected by the in situ surface measurements is
enhanced relative to the daytime, when the PBL is fully developed. For this reason, additional
Picarro measurements were made at night on August 13th 2015, when the PBL height is minimal.
Between 04:00 to 07:00 (LT), we performed Picarro measurements at different locations at
Chino, in order to map the different sources of $CH_4$ and verify that the large sources observed in
January, such as the lagoon, are still emitting in summer. Figure 11 shows the scatter plot of one
minute-average anomalies of $CH_4$ ($\Delta_{CH4}$) versus $CO_2$ ($\Delta_{CO2}$), colored by the $\delta^{13}C$ values,
measured by the Picarro on the night of August 13th 2015. During that night, the isotopic range
of $\delta^{13}C$ in sampled methane range from -45 ‰ to -65 ‰. These low $\delta^{13}C$ values are consistent
with the expectation that the sources of $CH_4$ in the Chino area are dominated by biogenic
emissions from dairy cows. In the feedlots (side triangles, Figure 11), $\Delta_{CH4}$ and $\Delta_{CO2}$ are well
correlated ($r^2$ = 0.90), because cows emit both gases (Kinsman et al., 1995). The observed
$\Delta_{CH4}/\Delta_{CO2}$ emission ratio, 48 ± 1.5 ppb/ppm, is in good agreement with a previous study



measuring this ratio from cow's breath (Lassen et al., 2012). Measurements obtained at less than
one meter away from cows (circles, Figure 11), had the lowest the $\delta^{13}C$ observed, ~-65 ‰, and
these points scale well with the linear correlation observed during the survey. This confirms that
the emission ratio derived surveying the feedlots is representative of biogenic emissions related
to enteric fermentation. Measurements obtained next to the lagoon (diamond marks, Figure 11),
the $^{12}CH_4$ concentrations enhanced by up to 40 ppm above background levels observed that
night, while the relative enhancement of $CO_2$ was much smaller. This extremely large $CH_4$
enhancement relative to $CO_2$ indicates a signature of $CH_4$ emissions from wet manure
management (lagoon), confirming that there is significant heterogeneity in the $CH_4$ sources
within the Chino dairy area.





5) Discussion
The fluxes derived by the FTS observations and the WRF-LES inversions, as well as previous
reported values are summarized in Table 1.
The top-down $CH_4$ estimate using FTS observations in Chino provide a range of fluxes from 1.4 to
4.8 ppt/s during January 2015 (Table 1), which are on the lower-end than previously published
estimates. These values of $CH_4$ flux estimates for January 2015 based on the FTS measurements
are consistent with the decrease in cows in Chino over the past several years as urbanization
spreads across the region.
Considering the decrease of dairy cows number by ~20% from 2010 to 2015, and using the
emission factor of 168 kg/yr per head (CARB 2015 inventory: enteric fermentation + dry manure
management), the $CH_4$ flux associated with dairy cows at Chino decreased from 2.0 to 1.7 ppt/s,
which agrees well with our low flux estimates derived from FTS observations. However, fluxes
derived using the simple mass balance approach differs from each other, exhibiting the
limitations of this method, even on a "golden day" (steady-state wind day on January 24[th]). The
WRF-LES inversions (Figures 10 and 12) and mobile in situ measurements with the Picarro
instrument (Figure 11) indicate that the $CH_4$ sources are not homogeneous within this local area.
In addition, wind measurements from the two local airports typically disagree regarding the
direction and speed (Figure 1, panels d, e, f, g, h, and i), and the WRF-LES tracer results indicate
non-homogeneous advection of tracers (Figure 8, right panels).
Figure 12 shows the map of the *a posteriori* X$_{CH4}$ fluxes (mean of January 15[th] and 16[th] runs) from
the WRF-LES simulations, superimposed on a Google earth map, with the location of dairy farms
represented by the red areas. The domain is decomposed into 16 boxes (Section 3.2), in which
the colors correspond to the *a posteriori* emissions derived from the WRF-LES inversions. Red
(blue) colors of a box mean more (less) $CH_4$ emissions compared to the *a priori* emissions, which
corresponds to the dairy cow emissions contained in the CARB 2015 inventory (emission factor
multiplied by the number of cows). Results of the inversion exhibit more $CH_4$ emissions at the
South and the Northeast parts of the domain, and emissions corresponding to dairy cows in the
center of the area.





The higher $CH_4$ emissions from the southwestern part of the domain can be attributed to the wet
manure lagoon (yellow pin, Figure 12) in January 2015. During the night of August 13[th] 2015,
Picarro measurements confirmed that the lagoon was still wet and emitted a considerable
amount of $CH_4$ relative to $CO_2$ (Figure 12). The second mobile Picarro instrument from JPL was
deployed on January 15[th] 2015 and measured $CH_4$ spikes up to 23 ppm near the wet manure
lagoon. The WRF-LES model also suggests higher methane fluxes in these regions (red boxes,
Figure 12). The CARB 2015 inventory estimates that manure management practices under wet
(e.g. lagoon) conditions emit more $CH_4$ than the dairy cows themselves: 187 kg $CH_4$ cow$^{-1}$ yr$^{-1}$
from wet manure management, 18 kg $CH_4$ cow$^{-1}$ yr$^{-1}$ from dry management practices, and 150 kg
$CH_4$ cow$^{-1}$ yr$^{-1}$ from enteric fermentation in the stomachs of dairy cows. Therefore, we expect
measurements in which the lagoon emissions were detected by our instruments will lead to
higher methane fluxes in the local region, compared to measurements detecting emissions from
enteric fermentation in cows alone. Bottom-up emission inventory of $CH_4$ is 2 times higher when
considering wet lagoons (Wennberg et al., 2012) instead of dry management practices (Peischl
et al., 2013) at Chino (Table1). The location and extent of wet lagoons in the Chino region is not
expected to be constant with time and could be altered due to changing land use and future
development in the area.  Bottom-up estimates of $CH_4$ emissions from dairies in the Chino region
could be further improved if the extent and location of wet manure lagoons were well-known.
The WRF-LES model also suggests higher methane fluxes in the Southeast (red boxes, Figure 13).
No dairy farms are located in these areas, but an inter-state pipelines is located nearby, thus
these $CH_4$ enhancements could be attributed to natural gas. The $^{13}CH_4$ Picarro measurements
indicate the Chino area is influenced by both fossil- and biogenic- related methane sources.  A
recent study has suggested the presence of considerable fugitive emissions of methane at Chino
(http://www.edf.org/climate/methanemaps/city-snapshots/los-angeles-area), probably due to
the advanced age of the pipelines. Natural gas leaks in the Chino area were not specifically
targeted during the time of this field campaign and cannot be confirmed using available data.
This possibility should thus be confirmed by future studies.
In addition to possible fugitive emissions at Chino, the inversion also predicts higher $CH_4$ flux in
the Northeastern region of the study domain, which is in the vicinity of a power plant that



reportedly emits a CH$_4$ flux roughly equivalent of one cow per year (only including enteric
fermentation)  (http://www.arb.ca.gov/cc/reporting/ghg-rep/reported_data/ghg-reports.htm).
Further analysis and measurements of fossil methane sources in the Chino area would help verify
potential contributions from fossil methane sources, including power plants and/or fugitive
natural gas pipeline emissions.
Overall, FTS and in situ Picarro measurements, as well as WRF-LES inversions, all demonstrate
that the CH$_4$ sources at Chino are heterogeneous, with a mixture of emissions from enteric
fermentation, wet and dry manure management practices, and possible additional fossil
methane emissions (from natural gas pipeline and power plants). The detection of CH$_4$ emissions
in the Chino region and discrepancies between top-down estimates could be further improved
with more FTS observations and concurrent in situ methane isotopes measurements combined
with high-resolution WRF-LES inversions. This would improve the spatial detection of the CH$_4$
emissions at Chino, in order to ameliorate the inventories among the individual sources in this
local area.





### 6) Summary and conclusions
In January 2015, four mobile low-resolution FTS (EM27/SUN) were deployed in a ~6 x 9 km area
in Chino (California), to assess $CH_4$ emissions related to dairy cows in the SoCAB farms. The
network of column measurements captured large spatial and temporal gradients of greenhouses
gases emitted from this small-scale area. Temporal variabilities of $X_{CH4}$ and $X_{CO2}$ can reach up to
20 ppb and 2 ppm, respectively, within less than a 10-minute interval with respect to wind
direction changes. This study demonstrate that these mobile FTS are therefore capable of
detecting local greenhouses gas signals and these measurements can be used to improve the
verification of $X_{CO2}$ and $X_{CH4}$ emissions at local scales.
Top-down estimates of $CH_4$ fluxes using the 2015 FTS observations in conjunction with wind
measurements are 1.4-4.8 ppt/s, which are in the low-end of the 2010 estimates (Peischl et al.,
2013), consistent with the decrease in cows in the Chino area. During this campaign, FTS
measurements were collected in close proximity to the sources (less than a few km) in order to
capture large signals from the local area. The main advantage of this type of deployment strategy
is to better constrain the emissions, while avoiding vertical mixing issues in the model with the
use of column measurements in the inversion. Therefore, the model transport errors, which
often limit the capacity of the model flux estimates, are considerably reduced. However, the close
proximity of the measurements to the sources makes the assumptions about homogeneity of the
sources and wind patterns questionable.
The FTS and the Picarro measurements detected various $CH_4$ signatures over Chino, with extreme
$CH_4$ enhancements measured nearby a wet lagoon (Picarro and FTS measurements enhanced by
40 ppm $CH_4$ and 60 ppb $X_{CH4}$, respectively) and possible fugitive fossil-related $CH_4$ emissions in
the area (indicated by higher $\delta^{13}C$ values than expected from biogenic emissions alone).
Wind speed and direction measurements derived from the two local airports (less than 10 km
apart), as well as the WRF meteorological simulations at different FTS sites, differ greatly with
each other, suggesting that an assumption of steady horizontal wind incorrect in the use of the
mass balance approach in our study. This may explain some discrepancy between the $CH_4$ flux
estimates from the mass balance approach and the Bayesian inversion.





This study demonstrates the value of using mobile column measurements for detection of local
$CH_4$ enhancements and the estimation of $CH_4$ emissions when these measurements are
combined with high-resolution modeling. High-resolution WRF-LES simulations were performed
on two dates, constrained by four column measurements each day, to map the heterogeneous
$CH_4$ sources at Chino. The average a posteriori flux over the domain is 3.2 ppt/s when only
considering the boxes in the center of the domain, and 4.7 ppt/s when all the boxes are averaged.
The major emitter (a wet manure lagoon) was identified by the inversion results, and is supported
by in-situ $^{13}CH_4$ measurements collected during the campaign. The $CH_4$ flux estimates are within
the range of the top-down mass balance emissions derived with the four FTS and estimates
reported by Peischl et al. 2013 (i.e., 2.1 to 6.5 ppt/s), showing that column measurements
combined with high resolution modeling can detect and possibly estimate $CH_4$ emissions.
The instrumental synergy (mobile in situ and column observations) coupled with a
comprehensive high-resolution model simulations allow estimation of local $CH_4$ fluxes, and can
be useful for improving emission inventories, especially in a complex megacity area, where the
different sources are often located within small areas.
This study highlights the complexity of estimating emissions at local scale when sources and wind
can exhibit heterogeneous patterns. Long term column observations and/or aircraft eddy
covariance measurements could improve estimations.
Acknowledgements:
The authors thank NASA and the W. M. Keck Institute for Space Studies for financial support.
MKD acknowledges NASA CMS support of the EM27/SUN deployment and LANL- LDRD
20110081DR for acquisition of the instrument. J. Chen, T. Jones, J. E. Franklin, and S. C. Wofsy
gratefully acknowledge funding provided by the National Science Foundation through MRI Award
1337512. January Campaign participants are Camille Viatte, Jacob Hedelius, Harrison Parker, Jia
Chen, Johnathan Franklin, Taylor Jones, Riley Duren, and Kristal Verhulst.





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

| study | time of study | sources | $CH_4$ emission (Gg/year) | $CH_4$ emission (ppt/s) |
|---|---|---|---|---|
| Peischl et al., 2013 | 2010 | inventory (dry manure + cows) | 28 | 2.5 |
| Peischl et al., 2013 | 2010 | aircraft measurements | 24-74 | 2.1-6.5 |
| Wennberg et al., 2012 | 2010 | inventory (wet manure + cows)* | 66 | 5.8 |
| CARB 2015 | 2015 | inventory (dry manure + cows) | 19 | 1.7 |
| This study | 2015 | FTS measurements only | 16-55 | 1.4-4.8 |
| This study | 2015 | WRF inversions | 36-54 | 3.2-4.7 |


* Value reported for the SoCAB, apportioned for Chino in this study.
Table1: Emissions of $CH_4$ at Chino.



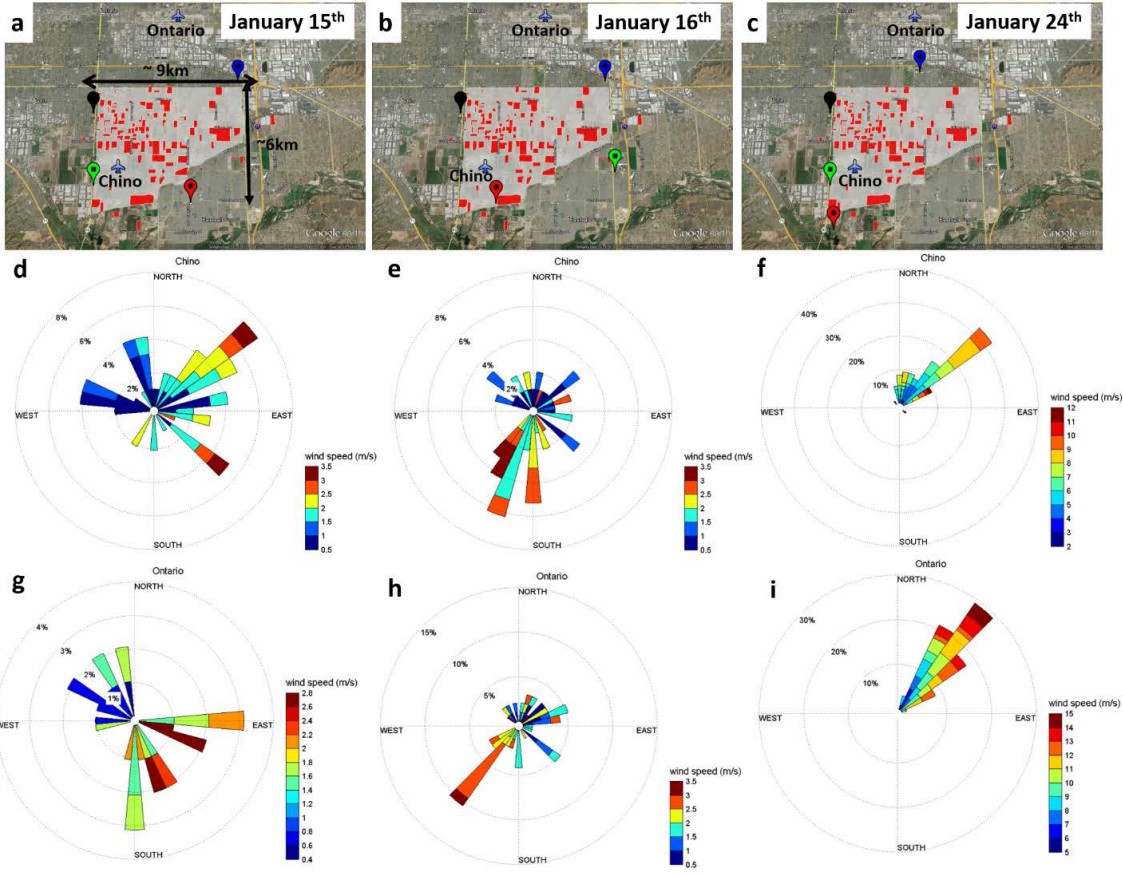


Figure 1: Three different days of measurements during the field campaign at Chino (~9 x 6 km) on the 15th,
16th, and 24th of January 2015. Upper panels (a, b, and c) show the chosen locations of the four EM27/SUN
(black, red, green, and blue pins correspond to the Caltech, LANL, Harvard1, and Harvard2 instruments,
respectively). The red marks on the map correspond to the dairy farms. Lower panels show wind roses of
ten-minute average of wind directions and wind speeds measured at the two local airports (at Chino on
panels d, e, and f, and at Ontario on panels g, h, and i). Map provided by GOOGLE EARTH V 7.1.2.2041, US
Dept. of State Geographer, Google, 2013, Image Landsat, Data SIO, NOAA, U.S, Navy, NGA, and GEBCO.





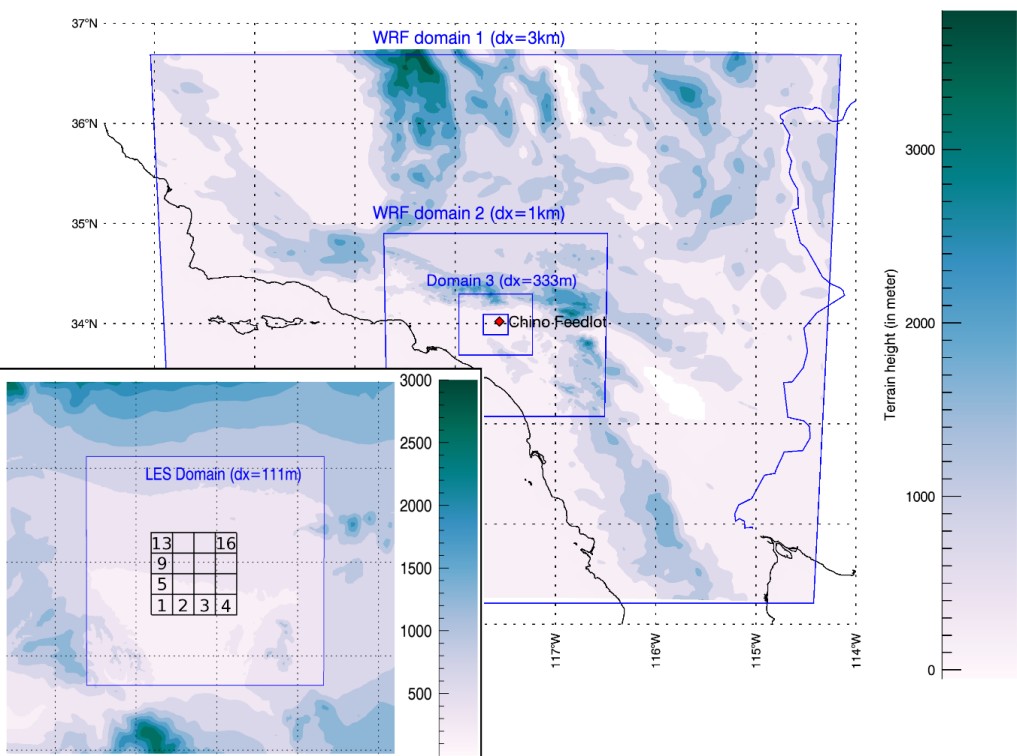


Figure 2: WRF-Chem simulation domains for the 4 grid resolutions (3-km; 1-km; 333-m; 111-m), with the
corresponding topography based on the Shuttle Radar Topographic Mission Digital Elevation Model at 90-
m resolution).  The 16 rectangular areas (2 x 2 km²) are shown on the LES domain map and numerate by
pixel numbers (Figure 10).



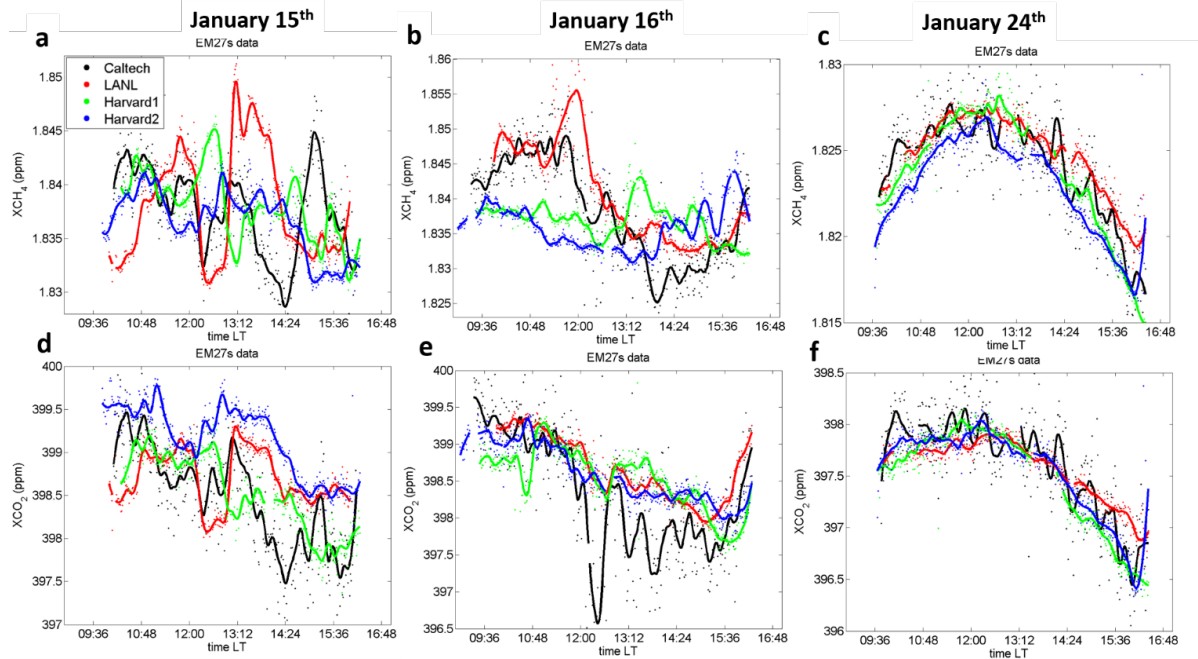


Figure 3: One minute-average time series of $X_{CH4}$ (upper panels a, b, and c) and $X_{CO2}$ (lower panels d, e, and
f) measured by the four EM27/SUN (black, red, green, and blue marks correspond to the Caltech, LANL,
Harvard1, and Harvard2 spectrometers, respectively).



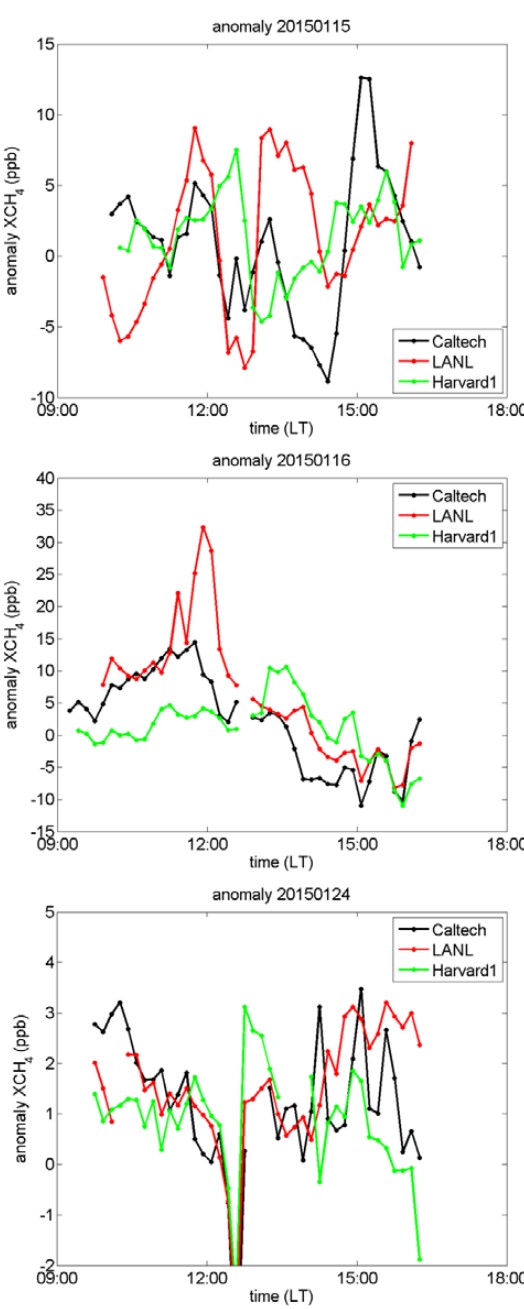


Figure 4: Time series of the 10-minute average $X_{CH4}$ anomaly ($\Delta_{XCH4}$, in ppb) computed relative to the
Harvard2 instrument for January 15th (upper panel), January 16th (middle panel), and on January 24th 2015
(lower panel).



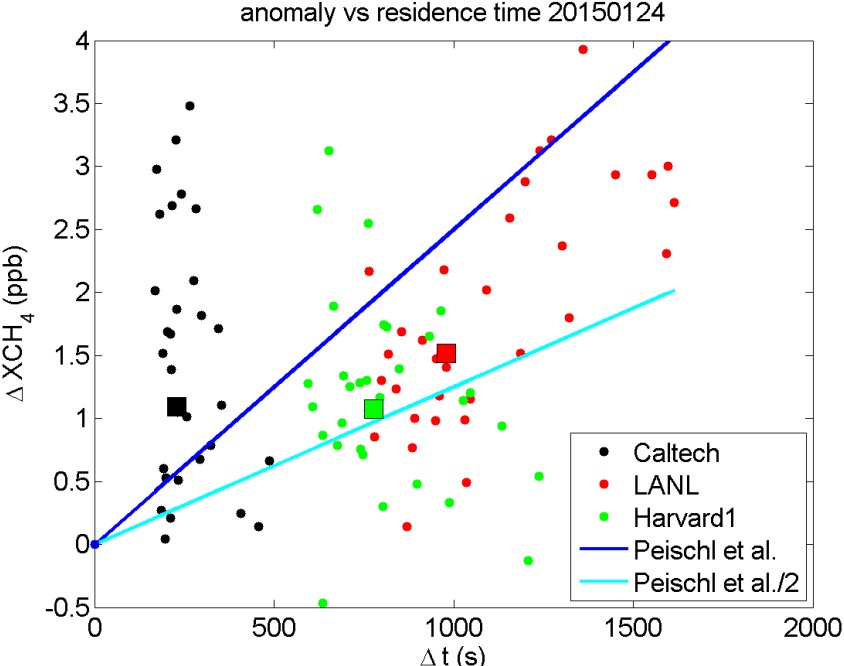


Figure 5: Estimated fluxes using FTS observations on January 24th. The 10-minute anomalies are plotted

against the time that airmass travelled over the dairies, so that the slopes are equivalent to $X_{CH4}$ fluxes (in

ppb/s, equation 5). The blue (and cyan) line represents the fluxes (and half of the value) estimated at

Chino in 2010 (Peischl et al., 2013). The squares are the medians of the data which correspond to the

estimated fluxes using the FTS observations (in black, red and green for the Caltech, LANL, and Harvard2

instruments).

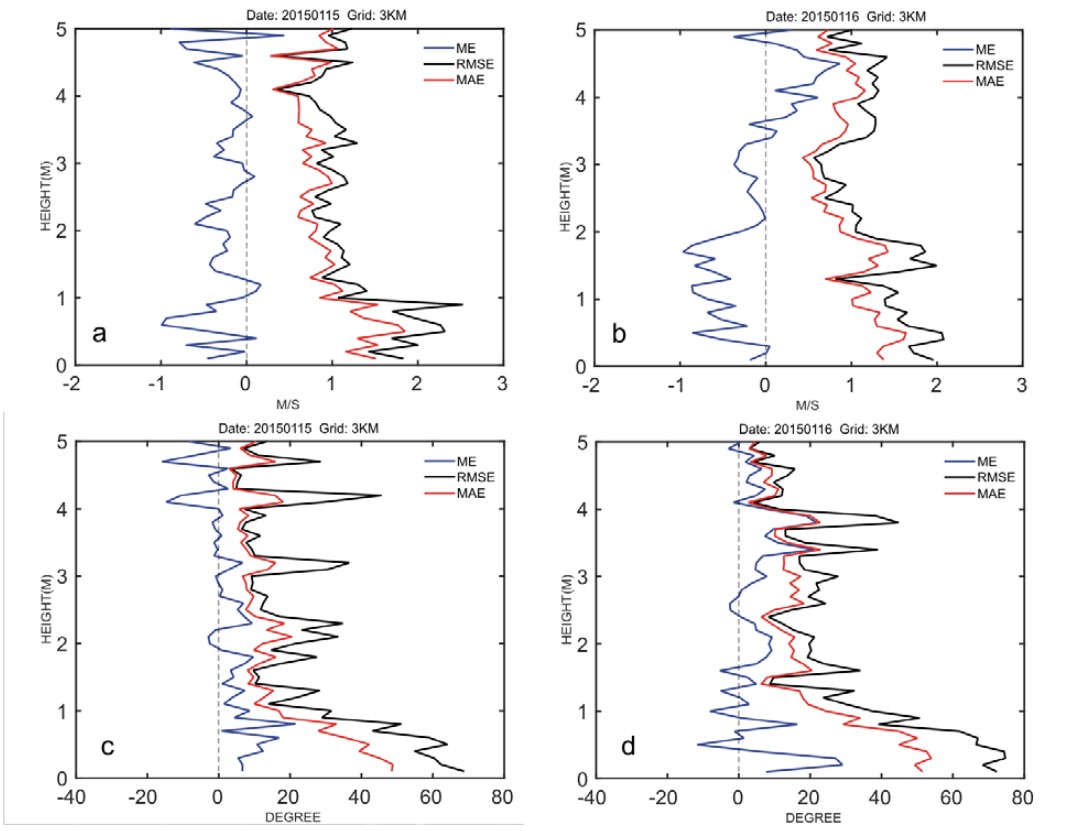

854

Figure 6: Vertical profiles of mean horizontal wind velocity errors (upper row) and direction (lower row)

averaged from the WMO radiosonde sites available across the 3-km domain, with the Mean Absolute

Error (in red), the Root Mean Square Error (in black), and the Mean Error (in blue). Only measurements

from 00z radiosondes were used in the evaluation.





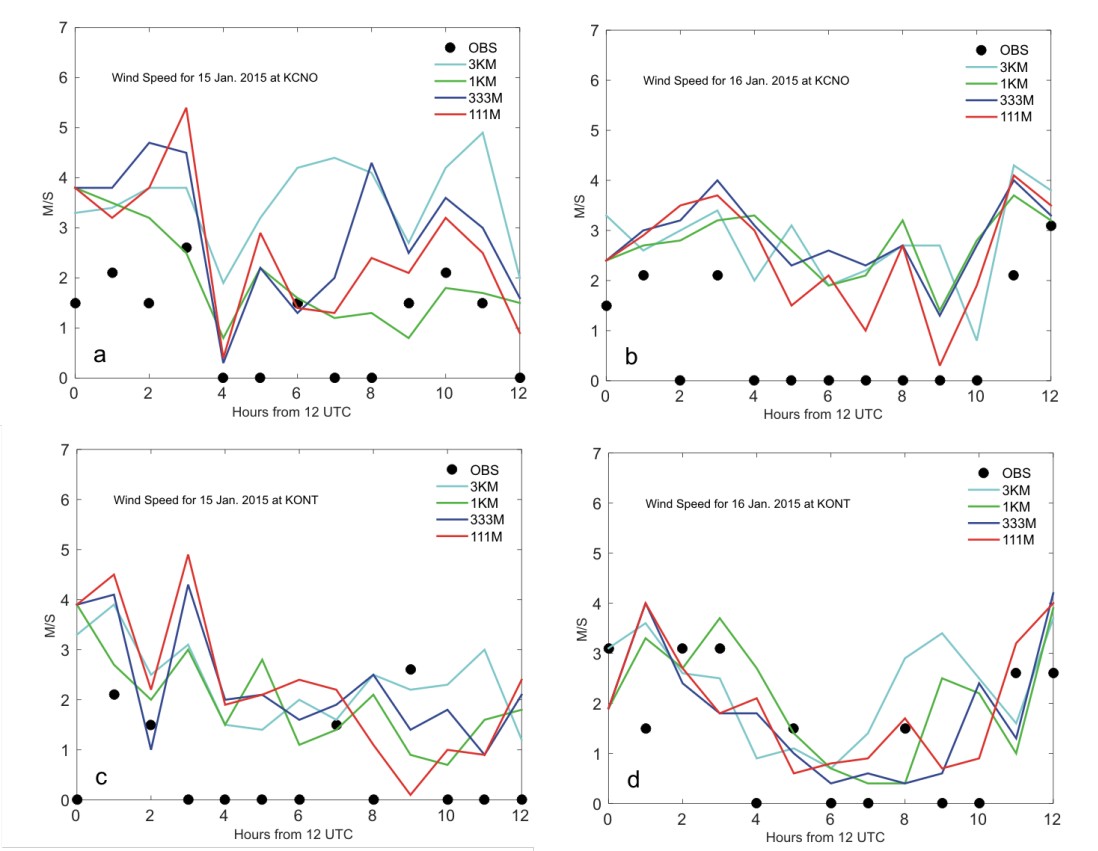


Figure 7: Mean horizontal 10-meter wind velocity in ms$^{-1}$ measured at Chino (KCNO) and Ontario (KONT)
airports for January 15$^{th}$ and 16$^{th}$ (black circles) compared to the simulated wind speed for different
resolutions using WRF hourly-averaged results. When black circles indicate zero, the wind velocity
measurements are below the WMO minimum threshold.




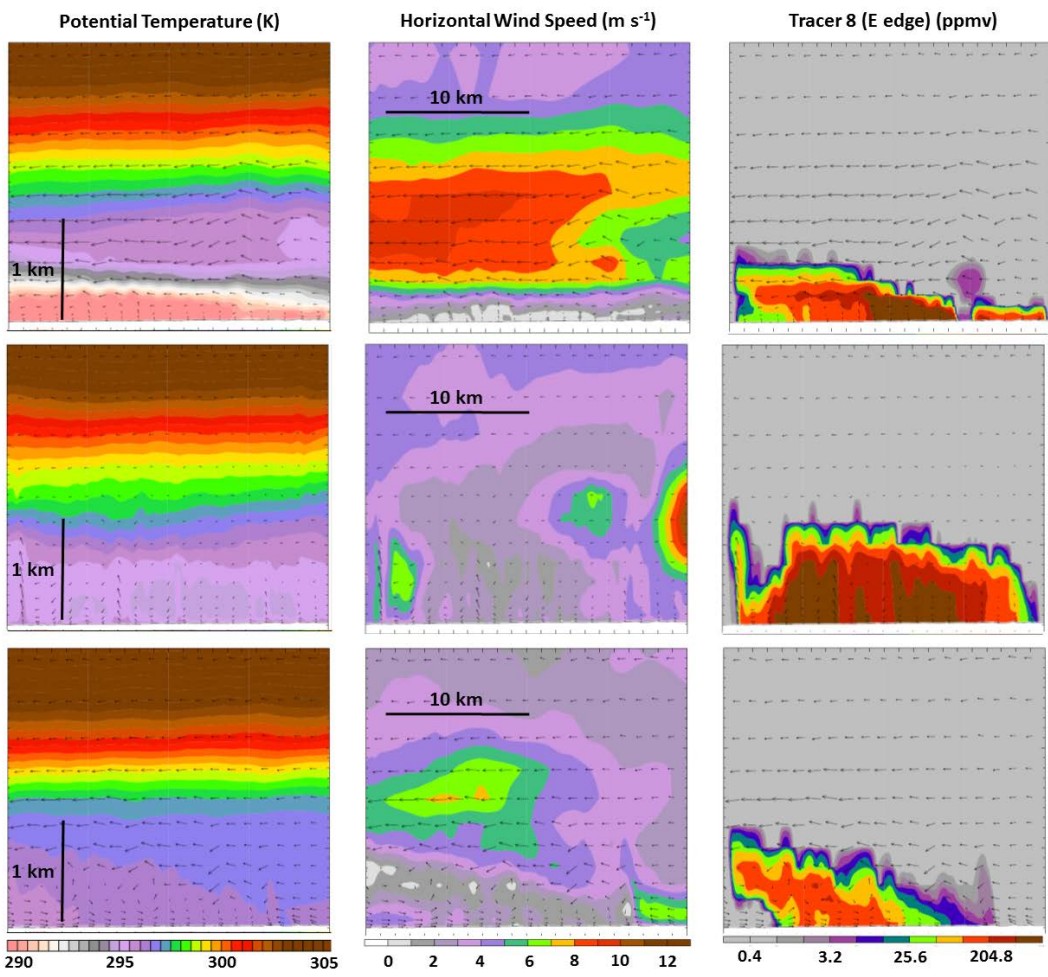


Figure 8: Vertical transects across the 111-m West-East WRF-LES simulation domain (pixels 5, 6, 7, and 8)
at 18:00 UTC of January 15th (upper row), 21:00 UTC (middle row), and 00:00 UTC (lower row). From left
to right, simulated data are shown for potential temperature (in K, left column), mean horizontal wind
speed and direction (in ms^-1 and degree, middle column), and passive tracer concentration released from
an eastern pixel of the emitting area (pixel 5, right column), to illustrate the relationship between the
three variables.



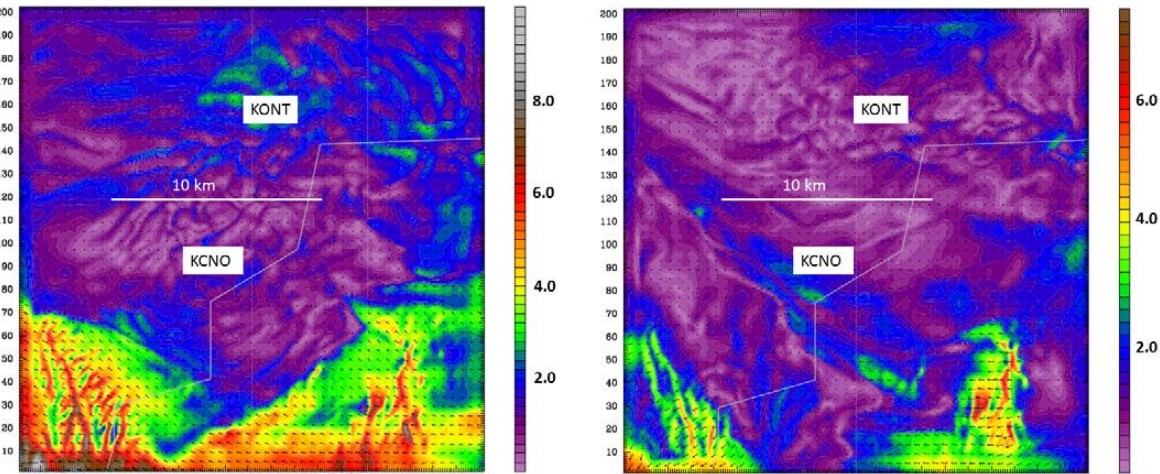


Figure 9: Mean horizontal wind field (in ms$^{-1}$) in the first level of the domain at 111-m resolution simulated by WRF-LES for January 15$^{th}$ (left panel), and January 16$^{th}$ 2015 (right panel), at 18:00 UTC. High wind speeds were simulated over the hills (southern part of the domain) whereas convective rolls, corresponding to organized turbulent eddies, are visible in the middle of the domain (i.e. over the feedlots of Chino), highlighting the importance of turbulent structures in representing the observed horizontal gradients of $CH_4$ concentrations. The locations of the Chino (KCNO) and Ontario (KONT) airports and the counties border (white line) are indicated.





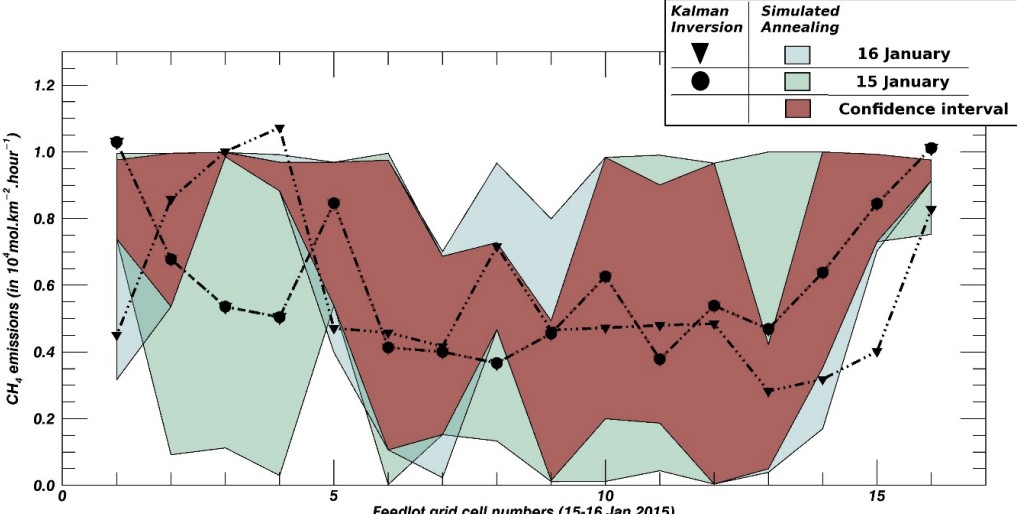

880

Figure 10: Emissions of $CH_4$ (in mol/km$^2$/hour) for the 16 pixels (2 x 2 km$^2$ shown In Figure 2) describing

the dairies for both days, i.e. the 15$^{th}$ and 16$^{th}$ of January 2015. The Bayesian mean emissions are shown

in black (triangles and circles) whereas the colored areas represent the accepted range of solutions using

the Simulated Annealing technique (see section 3.2).



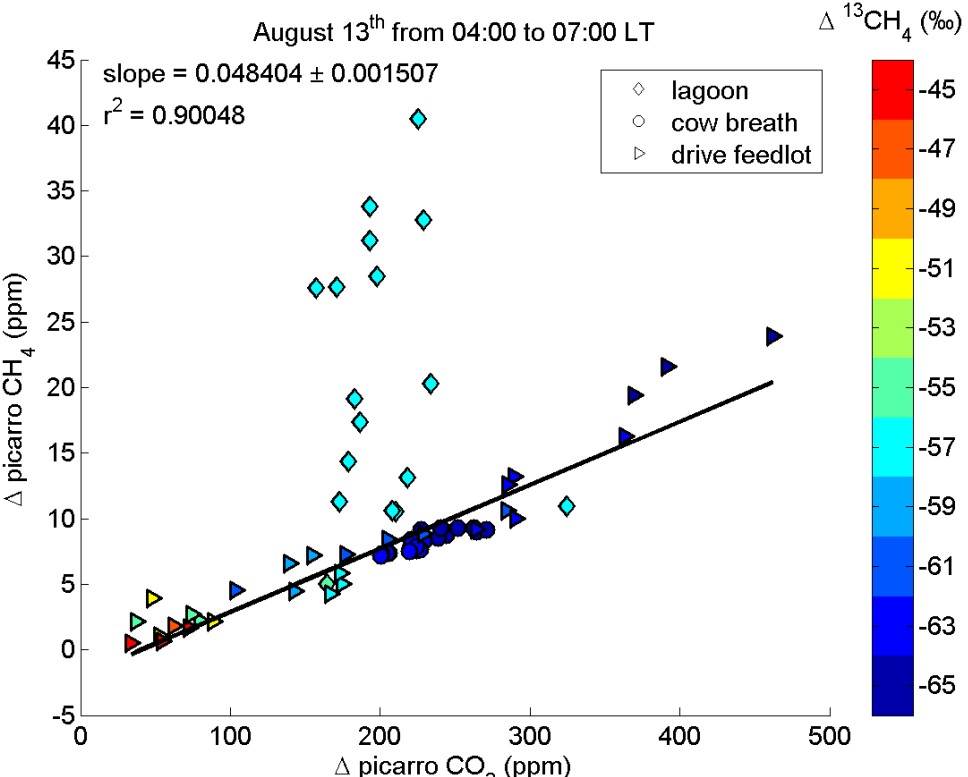


Figure 11: Scatter plot of one minute-average anomalies (from the 5 minutes smoothed) of CH₄ versus
CO₂, color coded by the delta CH4 values, measured by the Picarro on August 13th from 04:00 to 07:00
(LT).






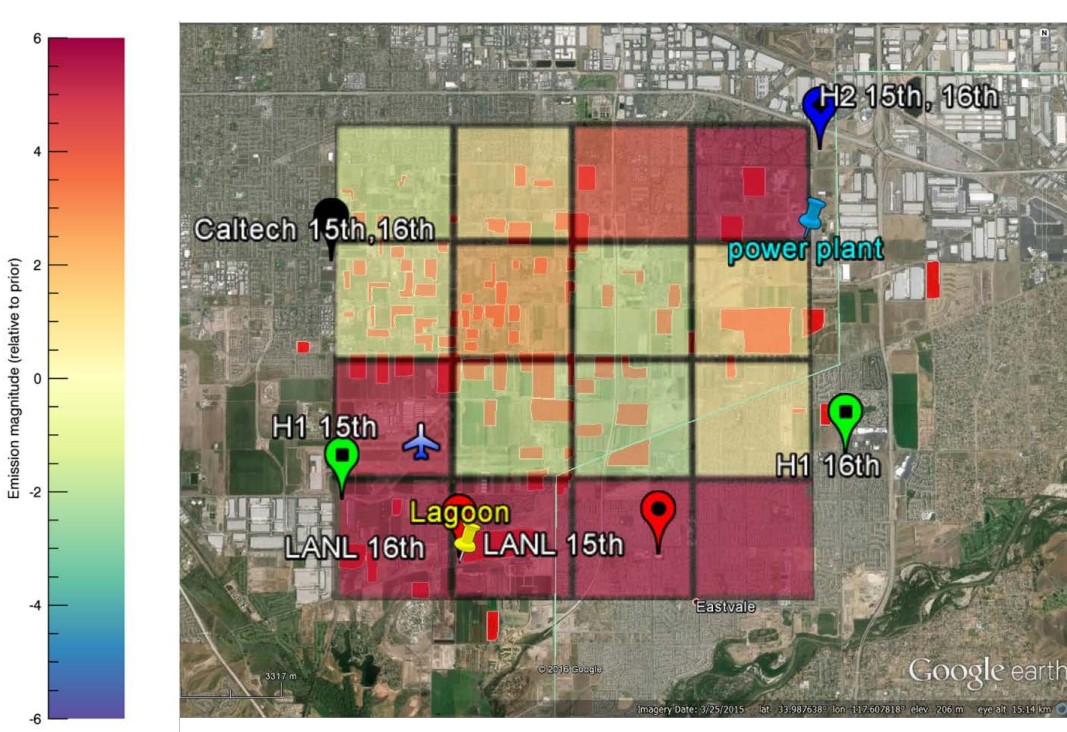

Figure 12: Map of the *a posteriori* X$_{CH4}$ fluxes (mean of January 15$^{th}$ and 16$^{th}$ runs) from the WRF-LES
simulations, superimposed on a Google earth map, where the dairy farms are represented by the red
areas as shown in Figure 1. The domain is decomposed in 16 boxes (2km x 2km), in which the colors
correspond to the *a posteriori* emissions from the WRF-LES inversions. Red (blue) colors mean more (less)
CH$_4$ emissions than dairy cows in that box. The locations of the lagoon (yellow pin) and the power plant
(blue pin) are also added on the map. Map provided by GOOGLE EARTH V 7.1.2.2041, US Dept. of State
Geographer, Google, 2013, Image Landsat, Data SIO, NOAA, U.S, Navy, NGA, and GEBCO.