# Peer review of "Methane emissions from dairies in the Los Angeles Basin"

_Atmospheric Chemistry and Physics, 2016_

## Referee Comment (RC1) · Anonymous Referee #1 · 8 Aug 2016

There is a growing interest in the use of remote sensing measurements for estimating strong local sources of methane. Research is needed to investigate methods for doing this, and their performance should be assessed using different types of measurements. This study has the right state-of-to-art ingredients to do this: A large local source of methane, total column as well is in in situ measurements and a model capable of resolving the relevant spatiotemporal scales. In the end, however, I'm left with the feeling that it is more important to the authors to convince the reader that this can be done, then to objectively assess the role of different factors influencing the outcome. As a result, and as I will explain further below, differences from the prior expectation are explained without taking proper account of methodological limitations. In addition, I had difficulty to understand some aspects of the method, which should have been explained better. Revisions in these directions are needed to make this study suitable

for publication.

**GENERAL POINTS**

It is known that the short-term regional scale variability in total column CH4 is dominated by local sources is well as the dynamics of weather systems. The model system that has been described takes the first into account, but not the second. I mean, it represents the weather but not the corresponding regional-scale patterns in XCH4. I was looking for ways in which the boundary conditions of CH4 where taken into account, but didn't find any. As a result, the fluxes that are derived may well account for unaccounted variations at the domain boundary. In Chen et al. an upwind-downwind approach was taken, which might indeed work under ideal conditions, with a well defined wind direction. As can be seen in Figure 4, this logic falls apart for low wind speeds. It would have been instructive to plot the zero line in this figure. Unlike the 24th of January, the gradient with respect to Harvard 2 is going everywhere, most notably on the 15th. Therefore I don't understand the role of Harvard 2 is a background on days with a low wind speed and doubt that it can be used that way. The authors may argue that variations in the background may be less important on days with lower wind speed, but this could en should then be demonstrated.

Before the WRF-LEF model is used to fit the FTS data it should be demonstrated that it has a reasonable skill in simulating the observed variability. The fit in S2 doesn't look great (what are the R2 values?), which makes me wonder about the comparison with the prior model. In this context it would be very useful to compare the use of WRF-LES with the use of WRF alone. What do we gain using LES? The results may not look great, but we'd learn about where the remaining problems are.

In line 370 it is mentioned that the modelled wind speed has an error of 'only' 1 m/s increasing further in the PBL. However, on the low wind-speed days this is a very substantial fraction of the total wind speed (according to Figure 1). What is the sensitivity of the emission estimation to errors in the modelling of wind speed and direction? I'd like
again to stress that it is important that we learn about what is critical in this approach. The size of the errors in wind speed and direction warrant closer inspection and a test of the potential impact.

It is clear to me why the mass balance approach is not used on the 15th and 16th of January. However, it is not clear why the WRF-LES method is not used on the 24th. This would provide a good opportunity to make a direct comparison between the two methods. It may not be the best case to demonstrate the added value of WRF LES, but as a general consistency check to include this comparison is nevertheless necessary.

Figures 4 and 5 confirm my worry about the emission pattern that is shown in Figure 12 and I wonder why it doesn't receive more attention. The pattern correlates with the configuration of the measurement 'network'. For an inversion the easiest way to fit the data is to modify the emissions in the same grid box as where the measurements are. It may either be that the measurements are very locally influenced, in which case they are not really representative of the domain that is being optimized. Otherwise the inversion may be trying to fit uncertain variables that are not part of the state vector, such as boundary conditions (see my earlier comment) or the emission distribution within 2x2km2 regions. With in situ measurements at LANL as high as 30 ppm CH4, this may well be an important factor. To me this sounds certainly like a more relevant factor to mention than the emission of a 1 cow per year power plant.

It was not clear to me why the simulated annealing approach was chosen. Since the inversion problem is linear, I don't see why its solution should be any different from the Bayesian method. If I understand well, the method was introduced to deal with the difficulty to define the B matrix. But how does simulated annealing solve this problem? Is it just efficiency at which different options for the B matrix can be tried out? If the greens functions are available, then I wonder how simulated annealing could be faster than a Bayesian inversion. This should be explained better.

SPECIFIC COMMENTS
Line 98: 'on a high-wind data'. What does this mean? Why is Chen 2016 not included in Table 1?

Line 117: In this sub section I am missing numbers for the estimated accuracy of the mobile column measurements. How realistic are the fits to the data derived later on in the light of these uncertainties?

Line 196: 'one way nested grids': S1 should provide further information about nesting. For CH4 the logical on-way nesting is from the small to the large domain. For meteorology, however, the reverse seems true. Some further explanation is needed.

Line 207: To avoid confusion about optimizing methane fluxes and optimizing meteorology it should be stated more clearly that 'data assimilation' is referring to the latter here. Line 255: What does the random draw refer to: the starting point of simulated annealing, the first guess, a random modification of the prior uncertainty? Does this modify the actual cost function that is optimized?

Line 257: 'Mean absolute error' This approach favors the setup that gives the largest freedom to the prior fluxes. This need not be the best solution, or maybe I do not understand what is meant here (see the previous point).

Line 290: 'C' io 'SC'?

Line 373: 'More relevant to this study' It is unclear why this should be the case. The difference between the two metrics represents a cancellation of wind speeds in calculating the mean. Such errors could still affect the estimated emissions, e.g. when winds with cancelling errors come from different directions.

Line 458: It is clear that the measurements in Figure 11 represent ruminant emissions. However, these samples are not representative of the air masses that are sampled with the FTS instruments. Therefore this result does not refer to the origin of the enhancements in total column CH4 discussed earlier. Confusion should be avoided on this point. **ACPD**
Line 579: 'The main advantage ...' This statement should be supported by evidence or a reference to other work.

Figure 1: The wind roses do not add up to 100%.

Figure 5: Please repeat that anomalies are relative to Harvard 2.

Figure 6: It should be made clear that these errors refer to errors in WRF-LES simulated wind speed and direction.

Figure 7: Mention the minimum WMO threshold value.

S1: How about the top boundary in the LES part of the domain?

TECHNICAL CORRECTIONS

Line 288: 'V is' io 'Vis'

Line 337: area of Chino in m2

Line 353: 'has' io 'have weaknesses'

Figure 8: Information on axes parameters and units should be given in the figures themselves instead of the caption. Please also add labels to the rows (as is done for columns)

Figure 9: Figure axes and legend without labels.

Title S1: 'models' io 'modes' Figures S4 and S5: Text is too small to read.

---

## Referee Comment (RC2) · Anonymous Referee #2 · 17 Nov 2016

Viatte et al. estimate methane emissions from a cluster of dairies located in Chino, California in the Los Angeles Basin. They measure methane column abundances using four mobile solar-viewing spectrometers situated about the cluster of dairies for four days in January 2015. They additionally use a Picarro analyzer to measure C12 and C13 methane from a mobile platform during the study, including one extra day in August 2015. These data are used to estimate emissions in two ways: (1) they calculate a crosswind flux using a mass balance approach, and (2) they use an inverse model. They find emissions between 1.4 and 4.8 ppt CH4/s from the cluster of dairies using the mass balance approach, and between 3.2 and 4.7 ppt CH4/s using the inversion technique. This emission estimate falls at the lower-end of previous estimates, which they credit to the declining number of dairies in the Chino area.

[Figure]

General comments

This is a novel method to determine emissions from a relatively confined area source. I recommend publication as long as revisions as described below are adequately provided.

Specific comments

1. How does the width of the FTS instruments' measurement swath compare to the width of the plume? One reason the "golden day" might not work for the basic flux method described here is because the three downwind FTS instruments are only measuring a small portion of the total plume. Please add some discussion of how this may affect the uncertainty of the flux method.

2. Line 455, Are these the results from a Keeling plot? Or are you just looking at the variability of d13C? If you look at the enhancements and a Keeling plot, do the d13C values make sense with the notion that the methane enhancements are consistent with dairy emissions?

3. How does the length of the measurement period couple into the uncertainties of the inversion? Are three days of measurements for this size and strength of emission sufficient? Could you have gotten away with fewer, or would more have helped?

4. Are there methane sources upwind that may contribute to the model placing methane emissions in the southeast portion of the study area? For example, how well does the LANL 16th site provide a background for the H1 16th site? If the H1 16th site sees an enhancement in methane that is not measured at the LANL 16th site, must the inverse model place those emissions in the southeastern most section of the grid?

5. Is there a way to quantify the reduction of dairies between 2010 and 2015? If so, I think this would be a good addition to help strengthen the conclusions based on comparisons with past emissions estimates.

Typos Line 20, 'a high-resolution atmospheric transport simulations' Line 60-61, 'a local

scales' Line 61, change 'apportion' to 'apportionment' Line 98, change 'on a high-wind data' Line 186, mismatch between 'surface emissions' and 'and its associated . . . tracers.' Line 256, change 'generate' to 'generated' to match case with 'iterated' Line 357, add 'the' before "golden day" Line 382, I think you mean 'slightly smaller than'? Line 542, change 'pipelines' to 'pipeline'

---

## Author Comment (AC1) · 11 Apr 2017

Response to anonymous referee #1 We would like to thanks the referees for their thorough comments. Note that our responses can be also found in the supplement.

General comments:

"It is known that the short-term regional scale variability in total column CH4 is dominated by local sources is well as the dynamics of weather systems. The model system that has been described takes the first into account, but not the second. I mean, it represents the weather but not the corresponding regional-scale patterns in XCH4. I was looking for ways in which the boundary conditions of CH4 where taken into account, but didn't find any. As a result, the fluxes that are derived may well account for unaccounted variations at the domain boundary."
In the inversion, we have represented the boundary conditions by selecting the observations of XCH4 dry air mole fractions unaffected by the local sources in our domain. These observations correspond to ideal wind conditions for each day assuming that over such a small domain, no spatial gradient in XCH4 mole fractions are expected and therefore one observation location is sufficient. The wind direction is changing but the minimum value is consistent over the two days at about 1.83-1.832ppm, except for a brief decrease to 1.825ppm on January 16th (for about 20 minutes). On January 15th, different instruments measure this value depending on the location relative to the wind direction. Two main reasons motivated our approach. First, we considered that the relative magnitude of the feedlot emissions is several orders of magnitude larger than any other sources in the area. The Chino feedlot is one of the largest sources of CH4 in the Los Angeles basin, which greatly simplifies the problem of the background conditions (i.e. the relative contributions from other sources are reduced). Nearby sources could still be significant if they are located at a very short distance from the sensors. But drive-around data and upwind EM27/SUN observations did not suggest any prevalent sources around the feedlot, as illustrated by the various EM27/SUN sensors measuring the same background at different times on January 15th. Second, if we were to consider a more regional approach to the problem, we would need a very accurate representation of CH4 mixing ratios (to few ppb) and therefore an accurate mapping of CH4 sources within several kilometers around the feedlot in order to simulate the signals from other contributors. This approach is likely to produce larger errors due to missing sources and incorrect magnitudes in the inventories as well as errors in the atmospheric transport, which could all introduce significant biases in the background mixing ratios (much larger than 2ppb). We would also need to define the inflow of CH4 from outside Los Angeles which would require another set of data outside the city limits. For these reasons, we decided to use an observation-based approach to avoid the complications due to model and inventory errors. We have modified the text to discuss the problem and explain more precisely our approach.

"In Chen et al. an upwind-downwind approach was taken, which might indeed work

under ideal conditions, with a well defined wind direction. As can be seen in Figure 4, this logic falls apart for low wind speeds. It would have been instructive to plot the zero line in this figure."

We agree and zero lines have been added to each plot.

"Unlike the 24th of January, the gradient with respect to Harvard 2 is going everywhere, most notably on the 15th. Therefore I don't understand the role of Harvard 2 is a background on days with a low wind speed and doubt that it can be used that way. The authors may argue that variations in the background may be less important on days with lower wind speed, but this could en should then be demonstrated."

We needed to have the same background instrument for all 3 days of measurements to test the mass balance approach. The Harvard2 XCH4 mean and standard deviation are the lowest of all the observations (Figure 3); which is why we have chosen this instrument as a measure of background conditions. We added a sentence on this limitation in the text.

"Before the WRF-LEF model is used to fit the FTS data it should be demonstrated that it has a reasonable skill in simulating the observed variability. The fit in S2 doesn't look great (what are the R2 values?), which makes me wonder about the comparison with the prior model. In this context it would be very useful to compare the use of WRF-LES with the use of WRF alone. What do we gain using LES? The results may not look great, but we'd learn about where the remaining problems are."

We have added a paragraph and a reference to a paper that will be published in the coming weeks (Gaudet et al., 2017). This study presents the ability of the Large Eddy Simulation mode to represent plume structures over short distances (less than 10km) compared to the mesoscale mode. Here, we have no simulation of the CH4 concentrations from coarse-resolution grids. Only 111-m concentrations were simulated.

"In line 370 it is mentioned that the modelled wind speed has an error of 'only' 1 m/s

increasing further in the PBL. However, on the low wind-speed days this is a very substantial fraction of the total wind speed (according to Figure 1). What is the sensitivity of the emission estimation to errors in the modelling of wind speed and direction? I'd like again to stress that it is important that we learn about what is critical in this approach. The size of the errors in wind speed and direction warrant closer inspection and a test of the potential impact."

The impact of transport errors on inverse estimates is an important topic that has been discussed in previous studies at lower resolutions (e.g. Lauvaux and Davis, 2014). In our study, it remains difficult to address this problem without the use of an ensemble of simulations. Instead, we decided to improve the transport using the WRF-FDDA system. We refer to another study (Deng et al., 2017) that will be published in the coming weeks. In this paper, we performed multiple simulations using WRF-FDDA and propagated the impact of the assimilation into the inverse fluxes. We showed that transport errors were significantly reduced when assimilating vertical profiles of meteorological observations. Here, our estimates of model errors, based on the chi2 normalized distance, suggest that transport errors are relatively small (less than 3ppb) allowing us to produce an inverse estimate for the area. We added a paragraph about model errors.

"It is clear to me why the mass balance approach is not used on the 15th and 16th of January. However, it is not clear why the WRF-LES method is not used on the 24th. This would provide a good opportunity to make a direct comparison between the two methods. It may not be the best case to demonstrate the added value of WRF LES, but as a general consistency check to include this comparison is nevertheless necessary."

We decided to avoid January 24th as the wind direction remains constant during the entire day. The inversion uses primarily the information from various wind conditions to constrain the distribution of the CH4 sources, similar to a triangulation approach. The results for January 15th and 16th revealed that sources are highly variables across the domain. Therefore, the lack of constraints on the spatial distribution (i.e. no change in

wind direction) would have been limiting on January 24th. We explain our choice in the section 4.3.1.

"Figures 4 and 5 confirm my worry about the emission pattern that is shown in Figure 12 and I wonder why it doesn't receive more attention. The pattern correlates with the configuration of the measurement 'network'. For an inversion the easiest way to fit the data is to modify the emissions in the same grid box as where the measurements are. It may either be that the measurements are very locally influenced, in which case they are not really representative of the domain that is being optimized. Otherwise the inversion may be trying to fit uncertain variables that are not part of the state vector, such as boundary conditions (see my earlier comment) or the emission distribution within 2x2km2 regions. With in situ measurements at LANL as high as 30 ppm CH4, this may well be an important factor. To me this sounds certainly like a more relevant factor to mention than the emission of a 1 cow per year power plant."

Figures S4 and S5 provide additional details to understand the attribution of signals in the state vector. Overall, the inversion is using the triangulation of information from various wind directions, which is why the two-day inversion is likely to produce the best estimate over the period. In general, with 2 sites, the flux corrections are applied to the nearby pixels. But this result is not valid when using 3 or 4 sensors. For example, on January 15th, two sensors (i.e. LANL and Caltech) show no correction in their vicinity whereas major changes are located near the other two sensors. The optimization is consistent across configurations of 3 or 4 sensors, detecting a major source in the North-East and the South-West, but not near the other two sensors. On January 16th, the fluxes remain unchanged near the Caltech instrument which demonstrates that the system is doing better than simply attributing the corrections to the nearly pixels. We provide additional comments based on Figure S4.

"It was not clear to me why the simulated annealing approach was chosen. Since the inversion problem is linear, I don't see why its solution should be any different from the Bayesian method. If I understand well, the method was introduced to deal with the

difficulty to define the B matrix. But how does simulated annealing solve this problem? Is it just efficiency at which different options for the B matrix can be tried out? If the greens functions are available, then I wonder how simulated annealing could be faster than a Bayesian inversion. This should be explained better."

We have described the Simulated Annealing (SA) in a separate paragraph. The main reason to perform the inversion in two steps is that SA has no prior information which means that if a pixel is not observed, the system will produce an infinite number of solutions. In addition, SA is producing scores (i.e. mismatches between observations and model concentrations) but no optimal solution is guaranteed. SA simply scans the space of solution (i.e. state space) without converging, whereas the Bayesian inversion will provide the optimal analytical solution considering our prescribed prior errors. In theory, if the optimal solution was found by the SA, then the two methods should agree, but there is no guarantee of such agreement.

Specific comments:

Line 98: 'on a high-wind data'. What does this mean? Why is Chen 2016 not included in Table 1?

This has be changed to 'recorded on favorable meteorological conditions (e. g. on a high-wind with constant direction)'. Chen et al. 2016 estimation has been added to Table 1.

Line 117: In this sub section I am missing numbers for the estimated accuracy of the mobile column measurements. How realistic are the fits to the data derived later on in the light of these uncertainties?

We have added a line about accuracy of these mobile column measurements: "Using Allan analysis, it has been found out that the precision of the differential column measurements ranges between 0.1-0.2 ppb with 10 min averaging time (Chen et al., 2016)".

Line 196: 'one way nested grids': S1 should provide further information about nesting. For CH4 the logical on-way nesting is from the small to the large domain. For meteorology, however, the reverse seems true. Some further explanation is needed.

"one-way" refers to the meteorology only. CH4 concentrations are simulated only in the 111-m grid. Considering the low risk that CH4 molecules would re-circulate over the area after having been advected out of the small domain, we ignored the coupling between the grids for CH4. This problem is more significant at larger scales when circulation of air masses is more circular around pressure centers or due to the terrain/surfaces. We provide additional details in S1.

Line 207: To avoid confusion about optimizing methane fluxes and optimizing meteorology it should be stated more clearly that 'data assimilation' is referring to the latter here.

This has been clarified. We have added "to optimize meteorological fields" after "Data assimilation".

Line 255: What does the random draw refer to: the starting point of simulated annealing, the first guess, a random modification of the prior uncertainty? Does this modify the actual cost function that is optimized?

Simulated Annealing is a random walk which implies that random draws of the prior emissions are made at every iteration. There is no optimization of a cost function in Simulated Annealing, only a score calculated for every possible solution. We clarified the sentence.

Line 257: 'Mean absolute error' This approach favors the setup that gives the largest freedom to the prior fluxes. This need not be the best solution, or maybe I do not understand what is meant here (see the previous point).

The impact of using the Mean Absolute Error is minor. We could have used the squared distances (like in a typical least-square regression analysis) or the Mean Error which

would account for source compensation.

Line 290: 'C' io 'SC'?

This has been changed

Line 373: 'More relevant to this study' It is unclear why this should be the case. The difference between the two metrics represents a cancellation of wind speeds in calculating the mean. Such errors could still affect the estimated emissions, e.g. when winds with cancelling errors come from different directions.

The sentence refers to the fact that the first error is computed for the Free Atmosphere (Free Troposphere) whereas our local enhancements are located in the PBL. We clarified the sentence.

Line 458: It is clear that the measurements in Figure 11 represent ruminant emissions. However, these samples are not representative of the air masses that are sampled with the FTS instruments. Therefore this result does not refer to the origin of the enhancements in total column CH4 discussed earlier. Confusion should be avoided on this point.

We have added 'measured by the Picarro instrument' to clarify this point.

Line 579: 'The main advantage ...' This statement should be supported by evidence or a reference to other work.

We have added a reference (Wunch et al. 2011).

Figure 1: The wind roses do not add up to 100%.

For clarity, we have not shown in this Figure wind data corresponding to null wind speeds, which is why the roses do not add up to 100%.

Figure 5: Please repeat that anomalies are relative to Harvard 2.

We have added this in the caption.

Figure 6: It should be made clear that these errors refer to errors in WRF-LES simulated wind speed and direction.

We have added 'Errors in WRF-LES simulated wind speed and direction:' at the beginning of the legend.

Figure 7: Mention the minimum WMO threshold value.

We added the value in the caption.

S1: How about the top boundary in the LES part of the domain?

The molecules of CH4 are not well-mixed over the PBL. For this reason, we avoided the use of PBL height as a relevant criterion in our calculation.

Technical corrections:

All of them have been changed Line 288: 'V is' io 'Vis' Line 337: area of Chino in m2 Line 353: 'has' io 'have weaknesses' Figure 8: Information on axes parameters and units should be given in the figures themselves instead of the caption. Please also add labels to the rows (as is done for columns) Figure 9: Figure axes and legend without labels. Title S1: 'models' io 'modes' Figures S4 and S5: Text is too small to read.

Please also note the supplement to this comment:
http://www.atmos-chem-phys-discuss.net/acp-2016-281/acp-2016-281-AC1-supplement.pdf

---

## Author Comment (AC2) · 11 Apr 2017

Response to anonymous referee #2 We would like to thanks the referees for their insightful comments. Note that responses can be also found in the supplement.

Specific comments:

1. How does the width of the FTS instruments' measurement swath compare to the width of the plume? One reason the "golden day" might not work for the basic flux method described here is because the three downwind FTS instruments are only measuring a small portion of the total plume. Please add some discussion of how this may affect the uncertainty of the flux method.

We have added in the text: 'For that day, the high wind speed causes a reduction of

the methane plume width across the feedlot, which may increase uncertainties on the mass-balance approach since the FTS' measurements may only detect a small portion of the total plume.'

2. Line 455, Are these the results from a Keeling plot? Or are you just looking at the variability of d13C? If you look at the enhancements and a Keeling plot, do the d13C values make sense with the notion that the methane enhancements are consistent with dairy emissions?

The values written at line 455 represent the variability of d13C measures in the feedlot. Looking at the Keeling plot for all the measurements, we obtained values around -40 ‰ (mixture of biogenic and fossil sources). However, when driving thought the feedlot with the Picarro instrument, methane enhancements associated with low d13C were recorded, especially when measuring right next to the cows. Therefore, methane emissions are found to be consistent with dairy emissions.

3. How does the length of the measurement period couple into the uncertainties of the inversion? Are three days of measurements for this size and strength of emission sufficient? Could you have gotten away with fewer, or would more have helped?

The assessment of uncertainties remains undetermined at this point and would require additional measurements to perform an independent evaluation. From the two days of data, we were able to retrieve a value that corresponds to previously published estimates. But the errors remain significant. Continuous measurements over several weeks would have provided a better estimate on the inverse fluxes by constraining the model with concentrations and meteorological fields for many different atmospheric conditions. However, results show (on Figure 10) that flux inversion obtained on two different days (the 15th and the 16th of January) lead to relatively similar results. Increasing the number of days in the inversions would reduce the confidence interval but the use of independent data (such as eddy-covariance flux tower measurements) would be needed to evaluate the performance of the system. The authors aim

to demonstrate that FTS measurement network coupled with WRF-LES inversion is a powerful methodology to derive local fluxes, even using relatively small number of data.

4. Are there methane sources upwind that may contribute to the model placing methane emissions in the southeast portion of the study area? For example, how well does the LANL 16th site provide a background for the H1 16th site? If the H1 16th site sees an enhancement in methane that is not measured at the LANL 16th site, must the inverse model place those emissions in the southeastern most section of the grid?

The source attribution problem is solved by combining various wind conditions over the one- or two-day time window (depending on the inversion case). As suggested in the question, information from observed enhancements will be attributed to specific areas depending on the downwind measurements site and the wind direction/speed. We selected two days with varying wind conditions in order to sample the entire domain at different times and with different combinations of sites (upwind/downwind). Whereas our sample size remains small, we have observed most of the wind directions (cf. wind roses in Figure 1) which suggest that our retrieved sources have been measured by more than one combination of sites. We agree though that some of our signals may be more constrained by a specific time (and therefore a specific set of sites) than others. A longer time window of observations with repeated enhancements from different wind directions would help support our findings. Here, we show that major sources can be identified across the feedlots. As an analogy, the inversion performs a triangulation of the sources assuming that our spatial coverage was sufficient. This coverage is a direct function of the wind direction variability as our sites are static.

5. Is there a way to quantify the reduction of dairies between 2010 and 2015? If so, I think this would be a good addition to help strengthen the conclusions based on comparisons with past emissions estimates.

Inventories of dairy and cow's numbers in this area are inexistent. Only previous studies within the area (Peischl et al. 2013, Wennberg et al. 2012) could be used to

quantify the reduction of dairies between 2010 and 2015.

Please also note the supplement to this comment:
http://www.atmos-chem-phys-discuss.net/acp-2016-281/acp-2016-281-AC2-supplement.pdf

---

## Author Response (AR2)

**Note to the editor**

We realized that the second part of the study, using the WRF-LES modeling system, was based on a previous version of the EM27 observations which have been further calibrated for surface pressure differences. Therefore, we have updated the results using the latest version of the data to make our analysis consistent throughout the paper. The emissions estimates have changed as presented in Table 1. The Figures 10, 12, S1, and S2 were also updated. Overall, our conclusions remain similar, with a better agreement compared to previous studies. The spatial distribution of the emissions remains mainly unchanged with small changes in the magnitude of the emissions.

We wanted to draw your attention to these changes and let you decide if they require additional reviews before publication.

Best regards,

Dr. Camille Viatte, on behlaf of all the co-authors.

**Response to referee #2**

**Somewhere, maybe in Figure 4, I suggest showing what the wind direction was for these measurements. Additionally, how did the authors treat negative anomalies? I didn't see discussion of that in the text.**

*As seen in Figures 1 and 8, wind measurements were collected at the two local airports, not co-located with our EM27 sensors. The mean horizontal wind direction and speed vary significantly as shown by the WRF-LES model, mostly due to the local topography. This spatial variability generates non-homogeneous advection of methane across the domain. Therefore, it is not obvious what wind directions data could be used to add to Figure 4. Also negative anomalies are treated as equal as positive anomalies, which could be improved in future studies by applying a more selective approach for the background $X_{CH4}$ conditions. We added some discussion in the Section 5 at lines 575-583.*

**I'm confused by the caption in Figure 12. What are units of emission magnitude? The caption text says this graph is showing the a posteriori fluxes, but the color legend says it is relative to the prior. Does this mean the prior has been subtracted from the posterior? If this is the difference, I suggest the authors show the total a posteriori emissions.**

*We show the multiplicative ratios between prior and posterior emissions. Because the prior emissions were kept constant over the domain, Figure 12 can be converted directly into a net emission map by using the initial value of the prior. We chose not to show the net posterior emissions due to large emission errors of individual pixels. Instead, we have used the ratio between the prior and the posterior emissions to highlight the spatial gradients instead of the net emissions which are highly uncertain. To help the readers do the conversion, we have added the following sentence in the caption: "A multiplicative ratio of 1 is equivalent to a flux of 2150 mol.km$^{-2}$.hour$^{-1}$."*

**In line 83, I would not classify the San Juan basin as a 'point source'.**

*We've delated the word "point".*

**In line 246, do you mean dairies are not distributed randomly across Chino? Or do you mean they are within your domain?**

*We assumed that the cows are randomly distributed within our domain for any given day, and therefore no error correlation should be assumed in our inversion.*

**In line 287 says 'we used the averaged of each day'. Averaged what? Lowest measured xCH4?**

*We clarified in the text how we used the daily minimum to define the background.*

*All the following revisions have been changed as suggested by the reviewer:*

**In line 60-61, change to 'flux estimations … are needed'**

**In line 62, change 'approach' to 'approaches'**

**In line 202, change to 'the effect of the most important'**

**In line 203, change to 'are not parameterized'**

**In line 286, add a significant digit to 1.830**

**In line 341, change to 'distance in meters'**

**In line 378, change to 'measurement days'**

**In the References section for Chen et al. (2016), there is an extra period after ACP abbreviation**